# Supervised spike sorting feasibility of noisy single-electrode extracellular recordings: Systematic study of human C-nociceptors recorded via microneurography

Alina Troglio[ID][1,2,3]*, Peter Konradi[4], Andrea Fiebig[1,2], Ariadna Pérez Garriga[ID][4], Rainer Röhrig[ID][4], James Dunham[5], Ekaterina Kutafina[ID][6,7°], Barbara Namer[1,3°]

**1** Research Group Neuroscience, Interdisciplinary Centre for Clinical Research (IZKF), Faculty of Medicine, RWTH Aachen University, Aachen, Germany, **2** Institute of Neurophysiology, RWTH Aachen University Hospital, Aachen, Germany, **3** Department of Anesthesiology, Intensive Care, Emergency and Pain Medicine, University Hospital Würzburg, Center for Interdisciplinary Pain Medicine, Würzburg, Germany, **4** Institute of Medical Informatics, Faculty of Medicine, RWTH Aachen University, Aachen, Germany, **5** Anaesthesia, Pain, and Critical Care Sciences, School of Physiology, Pharmacology & Neuroscience, University of Bristol, Bristol, United Kingdom, **6** Institute for Biomedical Informatics, Faculty of Medicine, University Hospital Cologne, University of Cologne, Cologne, Germany, **7** Scientific Center for Neuropathic Pain Research Aachen, SCN[AACHEN], Uniklinik RWTH Aachen University, Aachen, Germany

☙ These authors share the last authorship.
* alina.troglio@rwth-aachen.de

## Abstract

Sorting spikes from noisy single-channel in-vivo extracellular recordings is challenging, particularly due to the lack of ground truth data. Microneurography, an electrophysiological technique for studying peripheral sensory systems, employs experimental protocols that time-lock a subset of spikes. Stable propagation speed of nerve signals enables reliable sorting of these spikes. Leveraging this property, we established ground truth labels for data collected in two European laboratories and designed a proof-of-concept open-source pipeline to process data across diverse hardware and software systems. Using the labels derived from the time-locked spikes, we employed a supervised approach instead of the unsupervised methods typically used in spike sorting. We evaluated multiple low-dimensional representations of spikes and found that raw signal features outperformed more complex approaches, which are effective in brain recordings. However, the choice of the optimal features remained dataset-specific, influenced by the similarity of average spike shapes and the number of fibers contributing to the signal. Based on our findings, we recommend tailoring lightweight algorithms to individual recordings and assessing the "sortability feasibility" based on achieved accuracy and the research question before proceeding with sorting of non-time-locked spikes in future projects.

**Data availability statement:** The data sharing is restricted by the terms of the participant consent, which explicitly limits usage to to the scope of this research project. Therefore, these recordings are available only upon reasonable request from the corresponding author. All data used for plotting and statistical analyses are included within the Supporting information files. The computational pipeline code, visualization scripts, and a test data recording are available on GitHub https://github.com/Digital-C-Fiber/SpikeSortingPipeline. We have also used Zenodo to assign a DOI to the repository: 10.5281/zenodo.14552210.

**Funding:** This project was supported by a grant from the Interdisciplinary Center for Clinical Research within the faculty of Medicine at the RWTH Aachen University (NA2018-2024). BN is supported by the DFG (NA 970 6-2; NA 970 7-1; NA 970 9-2). The other authors received no specific funding for this work.

**Competing interests:** BN received consulting fees from Vertex. The other authors declare no competing interests. This does not alter our adherence to PLOS ONE policies on sharing data and materials.

## Introduction

Electrophysiological extracellular nerve recordings allow researchers to gain insight into the peripheral and central nervous system activity. In the peripheral nervous system, these recordings can capture crucial sensory information, such as object texture perception, motor action guidance, and warnings of potential tissue-damaging conditions [1–3]. The recorded signals often originate from multiple neurons, which are referred to as units. Accurate spike sorting is critical for analyzing the functionality of individual units. In some experimental setups, particularly for multi-channel *in-vitro* recordings, the sorting task can be efficiently handled due to the high quality of the recorded data and supplementary spatial resolution [4,5]. However, other essential techniques, such as single-electrode *in-vivo* microneurography experiments, present significant difficulties, including low signal-to-noise ratios, activity of the subject, and lack of spatial resolution via multielectrode arrays, which would provide more reliability through simultaneous recordings of spikes from multiple adjacent sites [6]. On the computational side, the lack of benchmark datasets with ground truth and variability in the experimental setups across internationally distributed labs restrict methodological development.

Nevertheless, a variety of methods have been proposed to address spike sorting in both single- and multi-channel contexts. Traditionally, features are extracted using principal component analysis (PCA) and then used for clustering [7,8]. In addition to this conventional approach, other methods include template matching in phase space [9–11], unsupervised Bayesian clustering algorithms [12,13], consensus-based clustering [14], support vector machine (SVM) approaches [15], and neural networks [16,17]. While these methods differ in their assumptions for the experimental setup and implementation, for example, some are suitable only for multi-electrode recordings or hardware-embedded, they all address the core problem of classifying spikes under noisy and spike overlapping conditions. Additionally, several software frameworks and toolkits have been designed to support spike sorting across a range of experimental settings to complement these algorithmic advancements. For instance, Spike2 (Cambridge Electronic Design Limited) offers comprehensive data acquisition and analysis solutions and SpikeInterface [18] provides an extensive spike sorting pipeline with several sorting algorithms, but an important point frequently omitted in experimental studies is the validation of sorting accuracy using ground truth data. While the algorithms perform well in experiments with clean recording conditions and excellent signal-to-noise ratios, applying them to the important use case of noisy *in-vivo* microneurography recordings can lead to unreliable results and false conclusions about neurophysiological processes [19]. Achieving the required recording quality during microneurography experiments with human chronic pain patients is highly time-consuming and yields a very low number of recorded nerve fibers, for example, typically only one fiber can be recorded during six hours of the patient remaining completely still. However, reliability is essential for analyzing single-neuron level discharge patterns to understand the sensory input and processes, such as synaptic transmission and signal encoding [20].

In the context of pain and itch research, these discharge patterns have been particularly underexplored. The simplistic paradigm "more spikes with higher frequency result in more pain sensation" is still the only one in active use [21]. To verify the hypothesis that different discharge patterns encode itch versus pain sensation within the same nerve fiber and to gain a deeper understanding of the underlying mechanisms of chronic pain and itch, it is essential to analyze and quantify these discharge patterns comprehensively, which requires spike sorting.

The electrophysiological technique of microneurography enables extracellular recordings from the small diameter unmyelinated nerve fibers that are crucial for signaling itch and pain. By obtaining these recordings in humans, including patients, it is possible to correlate neural responses with individual perceptions [22–24]. Microneurography captures single action potentials (spikes) from single peripheral nerve fibers. Previous studies, for example, have established a link between spontaneous activity in C-fibers and neuropathic pain in humans [25]. However, the single-contact electrode used in microneurography typically captures spikes from multiple nerve fibers within a single recording session, as C-fibers are anatomically clustered together in Remak-bundles. In addition, signal analysis is a challenge as the spikes from unmyelinated nerve fibers are small in comparison to the electrical noise.

This leads to simultaneous spike recordings from multiple nerve fibers along with noise from the subject's background physiological activity, such as spikes in sympathetic or highly temperature-sensitive nerve fibers. This combination of noise and multiple nerve fibers presents a significant challenge for spike sorting methods and, subsequently, for extracting meaningful discharge patterns.

Given the complexity of multi-fiber recordings and background noise, Forster and Handwerker have already attempted to solve these problems by introducing a spike sorting approach specifically for microneurography [26]. Their method relies on thresholding to detect spikes and works well when the signal-to-noise ratio is high. The algorithm generates "templates" of spike waveforms, which serve as references, in two phases: initially, templates are created based on detected spikes, and subsequently, it compares all spikes to these templates for sorting. While this method allows for clear visual association of spikes and removal of artifacts, it requires manual parameter adjustment and is very sensitive to signal quality. Despite the partial automatization of this approach introduced by Turnquist et al. [6], spike sorting in microneurography remains unreliable due to the complexity of spike morphologies, as spikes from different nerve fibers often have similar shapes, leading to misclassification. This issue is further complicated by variations in spike waveforms originating from a single nerve fiber caused by noise or changes in recording conditions [6].

To ensure robust microneurography studies, the "marking method" [27] was developed. It is a special stimulation protocol that time-locks a subset of spikes via electrical stimulation. It allows the experimenter to collect information about the number of fibers in the recording and to identify single-neuron firing patterns linked to electrical stimulation, providing insights into peripheral neural activity.

Specifically, the marking method leverages the consistent conduction velocity of unmyelinated nerve fibers when electrically stimulated at a low frequency, for example, 0.25 Hz (see Materials and Methods). In this manuscript, we refer to this type of stimulation as *background stimulation*. Spike detection and sorting are based on the latency of response relative to the background stimulus. We refer to a sequence of spikes resulting from a single fiber response to the background stimulus as *track.* These tracks are the equivalent of units in traditional spike sorting terminology. Typically, multiple tracks are visible when latency responses are displayed sequentially. We refer to this data representation as waterfall plots (see Materials and Methods). This facilitates the identification of C-fiber subtypes and the automatic sorting of spikes from different tracks. When additional spikes are evoked by *additional stimuli* applied between two background stimuli, the conduction speed of the nerve fiber slows down, and the latencies of subsequent spikes increase. This phenomenon is known as activity-dependent slowing (ADS) [28]. The magnitude of the slowing of speed correlates roughly with the number of previously elicited spikes. The sudden increase in response latency is called a "marking" by the *additional stimuli* on the *background stimulus*. Despite ADS, the tracks remain visible, and the spikes from the tracks can be reliably classified, allowing the study of the fiber behavior under various electrical stimulation protocols. However, to study the responses to

other significant stimulus types, for example, chemical and mechanical, as well as for a quantitative assessment of spontaneous activity observed in patients with peripheral neuropathies, an approach to classify all spikes accurately remains important.

In this work, we use the marking method to create ground truth datasets with reliable track labels to present a proof-of-concept computational pipeline for supervised spike sorting, focused on analyzing various feature sets and validation via experimentally available ground truth. We consider different data representations from simple (amplitude and width), through more sophisticated (spike sorting based on shape, phase, and distribution features (SS-SPDF) methods, as presented by Caro-Martín et al. [11]), to the raw waveform [29–31] and apply support vector machine (SVM) classification method to the per-subject classification task. When using the raw waveform, we deliberately avoid interpreting extracellularly recorded action potentials in terms of their canonical physiological phases (for example, threshold depolarization, rapid upstroke, or repolarization). Instead, we treat each waveform as a series of quantitative samples, abstracted and analyzed as data points, analogous to pixel intensities in automated image analysis algorithms. By analyzing spike morphologies and their impact on sorting accuracy, we aim to overcome the limitations of current unsupervised methods and develop strategies for improving spike sorting in microneurography that may be applicable to other neural data.

To assess generalization, we collected recordings from two laboratories employing different hardware and software configurations for microneurography. Due to the fact that the marking method is routinely used in microneurography, partial ground truth (i.e., background spikes) will be available for almost all experiments. We introduce, to the best of our knowledge, the first exploration of a supervised approach for microneurography data. While this analysis is limited to labeled (tracked) spikes, this work lays the foundation for extending supervised classification to untracked spikes, such as mechanically or chemically evoked activity. However, this future application will require robust detection strategies and validation techniques, which we identify as key next steps and ongoing work. Nevertheless, by assessing the classifier's accuracy on the tracked background spikes, we can estimate its ability to reliably sort the untracked spikes despite having ground truth only for the tracked data.

To support this analysis, we developed a data infrastructure, built for harmonizing and analyzing microneurography data, including a metadata standard [32] tailored for microneurography experiments employing odML and odML-tables [33,34], a Python library to export data from a data acquisition system [35], and openMNGlab [36], an open-source analytical framework (in development). This infrastructure allowed us to utilize 26 recordings from microneurography laboratories in Aachen, Germany (datasets labeled with A) and Bristol, United Kingdom (datasets labeled with B).

Our pipeline is the first step to efficiently and reliably analyze rare and valuable patient-derived recordings to better understand and treat chronic pain and itch. Further, it can serve as a "guideline" for testing and adapting spike sorting methods based on spike feature sets in diverse neuro-electrophysiological datasets. Additionally, the marking method provides a practical example of how to generate ground truth data with minimal human effort. By systematically analyzing microneurography recordings, we were able to characterize the morphological variability of spike shapes, providing important insights into the expected limitations and performance of spike sorting approaches in this context.

## Results

### Open-source spike sorting pipeline for microneurography data

We designed and evaluated a pipeline for the automated and systematic evaluation of spike sorting in microneurographic recordings [37]. This pipeline enables subsequent comparison of various feature extraction methods, as inputs for supervised machine learning models for microneurography data. To ensure the reproducibility and scalability of our analysis, we implemented the entire processing pipeline using Snakemake [38], a workflow management system designed for robust and automated data analysis. This framework allows us to define each step, from raw data reading and preprocessing to feature set extraction and classification, as modular, trackable processes. By leveraging Snakemake, we were able to process data from multiple sources efficiently and maintain a consistent computational environment across experiments.

We pre-analyze microneurography recordings using a 'waterfall' representation, which facilitates visualization of spike time alignment to tracks during low-frequency stimulation (Fig 1). This vertical alignment occurs because C-nociceptors have a consistent conduction velocity when stimulated electrically at a fixed low frequency. The alignment enables the identification and labeling of spike events by individual nerve fibers with a track number. In Fig 1, two tracks, Track1 (purple) and Track2 (orange), are identified from a nerve fascicle. Spikes are extracted within a 3 ms window from the signal, and the raw signal, spike and stimulation onsets, and track labels are combined into a harmonized NIX file [39]. Data harmonization is essential because different data acquisition systems are used across the two labs. We extract various feature sets from the spike waveforms. These feature sets (in short: simple, SPDF, W), which are summarized in the table within Fig 1 and further detailed in the Materials and Methods section, allow us to investigate which feature set works best for sorting.

To assess the sorting accuracy on the tracked spikes, we employ a 5-fold cross-validation approach, splitting the labeled spike data into 80% training and 20% testing sets. Evaluation metrics are averaged across all folds to provide a more general performance score.

A Support Vector Machine (SVM) with a radial basis function (RBF) kernel is used as the classifier for its transparency, computational efficiency, and low number of hyperparameters. We calculate key performance metrics for sorting evaluation, including accuracy, precision, recall, and the macro-averaged F1-score (data in Supporting Information, S1–S4 Files).

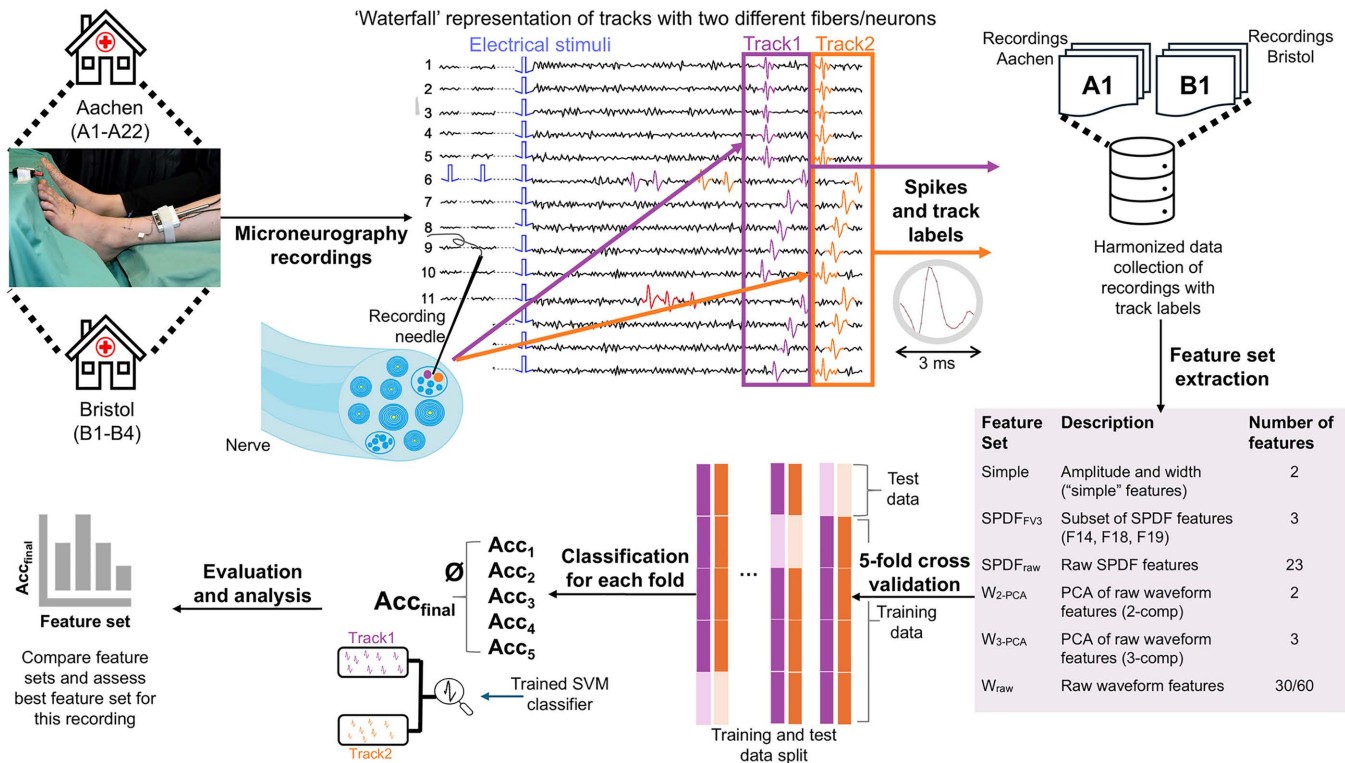

**Fig 1. Overview of the pipeline designed to sort spikes in microneurographic recordings.** Data is collected from laboratories in Aachen and Bristol and pre-analyzed using a 'waterfall' representation to align spikes based on C-fiber conduction speed. Two example *tracks* (purple and orange) are shown. The raw signal and extracted spike waveforms are harmonized, and multiple feature sets are computed (see Table) to assess their suitability for sorting. A Support Vector Machine (SVM) classifier with a radial basis function (RBF) kernel was applied to classify spikes, using 5-fold cross-validation and standard evaluation metrics (accuracy, precision, recall, and F1-score).

## Raw spike waveform ($W_{raw}$) is the best input feature set for sorting spikes in microneurography recordings

We computed the averaged accuracy across all five cross-validation folds and tested it on six feature sets (Fig 2A). As feature sets, we included amplitude and width (simple), two feature sets derived from the SS-SPDF method ($SPDF_{FV3}$ and $SPDF_{raw}$), and three feature sets derived from the raw waveform ($W_{2-PCA}$, $W_{3-PCA}$, and $W_{raw}$). Details of the feature set computations are provided in the Materials and Methods section. The $W_{raw}$ feature set has the highest mean accuracy, while the simple feature set has the lowest mean. The mean accuracy of the simple feature set is 0.59, compared to 0.73 for $W_{raw}$ with the other feature sets falling in between. To further explore and substantiate these findings, we conducted a statistical analysis. The Wilcoxon signed-rank test with Bonferroni correction for multiple comparisons revealed no significant differences between the accuracies of most feature sets (Bonferroni-adjusted threshold $\alpha_{adj} = 0.01/n \approx 0.00067$, where $n = 15$ pairwise comparisons). Nevertheless, $W_{raw}$ performs significantly better than all other feature sets (Table 1).

Although $W_{raw}$ has the highest accuracy mean and median across all feature sets (Figs 2A, B), certain datasets exhibit superior performance with other feature sets, for example, dataset A1 performs best with the $SPDF_{FV3}$ feature vector as input. Therefore, evaluating the optimal feature set for each specific dataset is essential.

## Template morphological similarity and fiber count as indicators of sorting success

In Fig 3, we present exemplary plots for datasets A1 and A6, showing all tracked spikes and a corresponding *template* for each fiber in the recording, generated by averaging all spikes within each track. Spike templates for all recordings are provided in the Supporting Information (S5 File). To quantify the impact of similar spike shapes on sorting accuracy we used three distance metrics (see Materials and Methods), including root mean square error (RMSE), to evaluate the relationship between the visual similarity of templates and its impact on sorting accuracy (Fig 4A). For instance, the RMSE between templates in dataset A1 is 1.20 (max accuracy 0.97), while in dataset A6 the RMSE is significantly lower at 0.21 (accuracy 0.75) indicating higher similarity between templates and, consequently, lower sorting accuracy. This analysis demonstrates that lower distance scores often correspond to poorer sorting results due to template overlap, suggesting that template similarity is a key factor in assessing sorting quality. Additionally, distance metrics may serve as pre-sorting indicators to identify recordings that may not meet the requirements for good sorting outcomes.

Fig 4B shows the mean classification accuracy across all feature sets with different marker colors representing the number of fibers or tracks (2–6) in each recording. As expected, recordings with fewer fibers (e.g., two fibers) consistently achieve higher classification accuracy. In contrast, the classification accuracy tends to decrease as the number of fibers increases, indicating the challenge of differentiating between more similar spikes. Beyond fiber count, the feature set choice also impacts classification performance and aligns with the previously presented results.

Additionally, random chance accuracies (dashed lines) for each fiber count provide a reference for judging sorting quality. Recordings with mean accuracies close to or below these thresholds indicate poor separation of fiber classes or high template similarity. This baseline comparison allows us to identify recordings that are inherently challenging to classify accurately, even with optimal feature sets.

## Limitations of unsupervised clustering with PCA features for microneurography data

As mentioned in the Introduction, the traditional spike sorting pipeline after spike detection often involves extracting principal component (PCA) features and evaluating cluster separability to differentiate fiber classes [8,40]. This approach assumes that unsupervised clustering can effectively differentiate between classes by relying on the natural separation of clusters in the PCA feature space. However, our findings suggest that this approach is most of the time unsuitable for microneurography data.

For example, when analyzing A1, we observe a high classification accuracy (0.97), yet no clear cluster separation when visualized in either two-dimensional (Fig 5A) or three-dimensional PCA space (Fig 5B). This discrepancy highlights a key

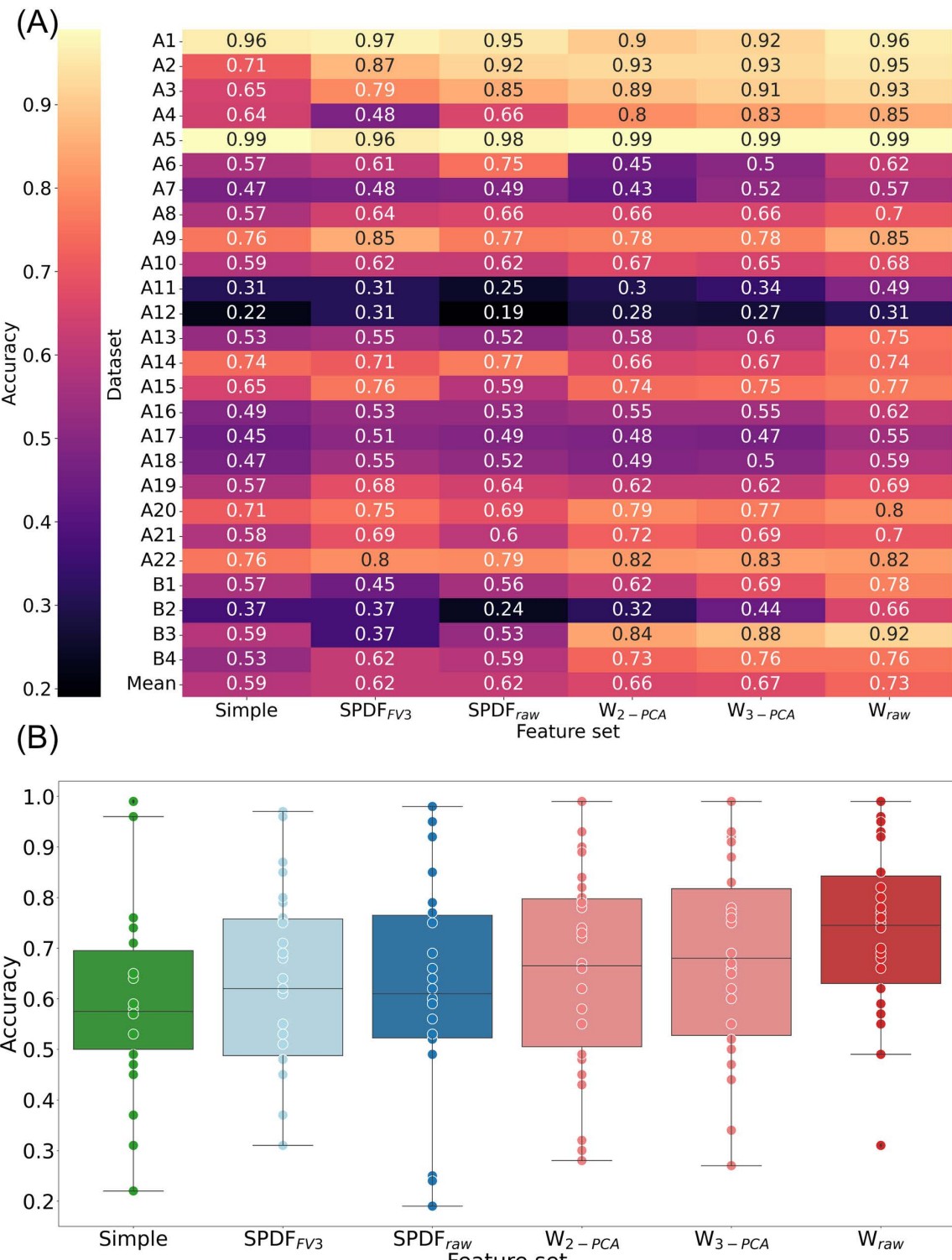

**Fig 2. Accuracy results for all datasets grouped by feature set.** (A) Individual results for each dataset. Darker shades represent lower scores, while lighter shades indicate higher scores. For $W_{raw}$, the mean accuracy is the highest, with 0.73. (B) Distribution of accuracies for each feature set. Boxes are drawn from the first quartile (median of the lower half of the score distribution) to the third quartile (median of the upper half of the score distribution). The line in the box marks the median of the scores. The lower and upper whiskers are bounded by the 1.5 interquartile range (IQR), which is the distance

between the first and third quartiles. Scores outside of the 1.5 IQR bound are plotted as outliers. The dots mark the scores for individual datasets. Each color represents a different feature set group, with lighter, more transparent shades indicating the subset of the SS-SPDF feature vector or PCA in two and three components of the raw signal shown in the brighter color. Green stands for the simple feature set, blue for the feature sets of the SPDF method, and red for the raw waveform feature sets.

**Table 1. Results of the Wilcoxon signed-rank test comparing accuracy differences across all feature sets. Each row represents a pairwise comparison between one feature set and the others, displaying the statistical significance based on the Bonferroni-corrected alpha level ($\alpha_{adj} = 0.01/15 \approx 0.00067$). Red: not significant; green: p < $\alpha_{adj}$.**

| | Simple | SPDF$_{FV3}$ | SPDF$_{raw}$ | W$_{2\text{-PCA}}$ | W$_{3\text{-PCA}}$ | W$_{raw}$ |
|---|---|---|---|---|---|---|
| **Simple** | | | | | | |
| **SPDF$_{FV3}$** | 0.0356 | | | | | |
| **SPDF$_{raw}$** | 0.0890 | 0.6269 | | | | |
| **W$_{2\text{-PCA}}$** | 0.0043 | 0.6349 | 0.0346 | | | |
| **W$_{3\text{-PCA}}$** | 0.0004 | 0.1963 | 0.0097 | 0.0126 | | |
| **W$_{raw}$** | $2.667 \times 10^{-5}$ | $2.808 \times 10^{-5}$ | $1.907 \times 10^{-5}$ | $3.01 \times 10^{-5}$ | $2.171 \times 10^{-5}$ | |

limitation: visual inspection of PCA projections may incorrectly suggest a single track when multiple tracks are present. To further test this, we applied k-means clustering and evaluated performance using ground-truth labels.

While PCA is typically limited to the first 2–3 components in practice, we extended the analysis to include up to eight components to ensure we did not overlook higher-dimensional separability. Nonetheless, clustering performance metrics, including Adjusted Rand Index (ARI), Normalized Mutual Information (NMI), and V-measure (defined and implemented in the scikit-learn library [41]), remained low and stable across all component counts (Table 2). For example, even with five components capturing 82% of the variance (cumulative explained variance for A1 and A3 is visualized in S6 File), the V-measure peaked at only 0.62, indicating that clustering performance remained poor despite higher-dimensional feature representations.

All clustering results using k-means with the presented feature sets are reported in the Supporting Information (S7–S9 Files). Interestingly, when analyzing the SPDF-derived feature set (SPDF$_{FV3}$), it revealed two clearly separated clusters (Fig 5C) for A1, which aligned with the known tracks and coincided with a high ARI score (S7, 0.88). However, it is important to note that the number of clusters (k) was fixed to match the known number of tracks, which introduces a bias that artificially inflates clustering performance. Thus, the reported metrics should be interpreted with caution.

In contrast, for dataset A3, despite high classification scores for the W$_{raw}$ feature set (0.93), no visually distinct clusters were observed in any of the feature sets (Fig 5D–F). This further shows the disconnect between classification accuracy and the visibility of separable clusters in feature space, especially under unsupervised assumptions.

Considering these findings, we note that both PCA- and SPDF-based feature sets perform well in terms of supervised classification accuracy. Additionally, SPDF features can provide more robust clustering behavior and better reflect underlying track separations, particularly in cases like A1.

However, these results underscore the need for caution among microneurography researchers, as relying solely on unsupervised methods like k-means clustering applied to PCA feature sets may yield unreliable results in this context. We advocate supervised methods that can leverage prior knowledge to improve classification accuracy when group boundaries are not inherently clear as unsupervised methods may suggest more or fewer classes than exist.

## Limitations of unsupervised spike sorting algorithm applying SpikeInterface

To further evaluate the limitations of unsupervised spike sorting pipelines for microneurography data, we applied the SpikeInterface framework [18] to A1 and A5, using the NIX format for seamless integration, as they had the best

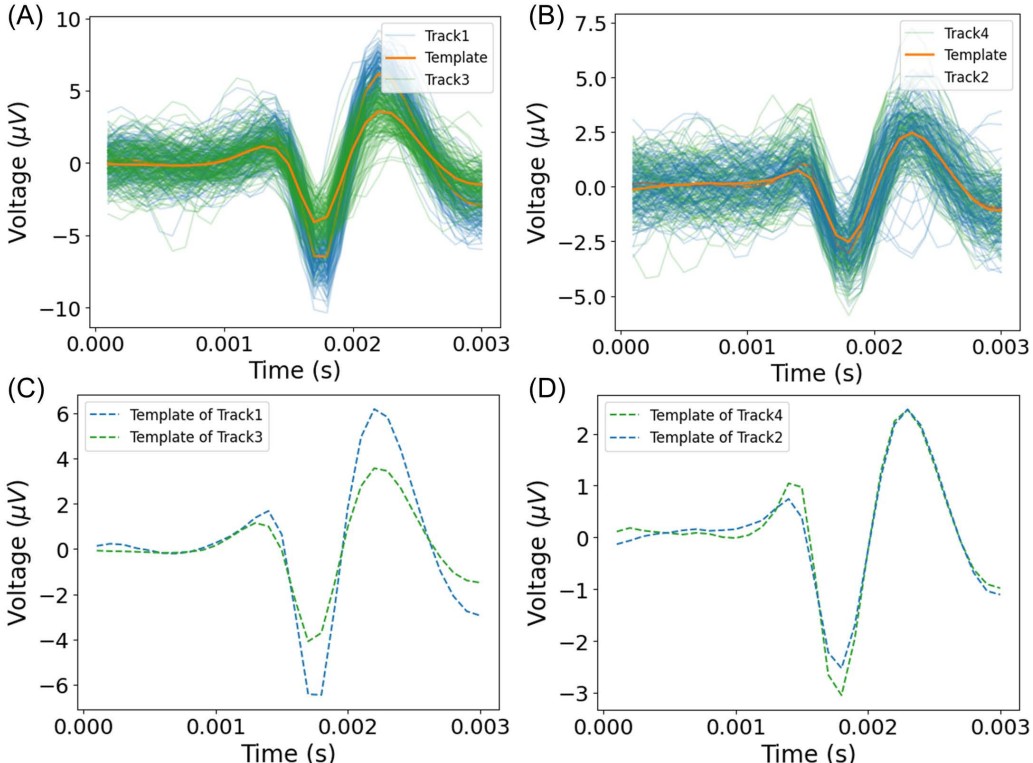

**Fig 3. Spike waveforms and corresponding templates for datasets A1 and A6 were created by averaging all detected spikes.** (A) Waveforms for both tracks in recording A1 show the blue track with a higher amplitude than the green track. (B) Waveforms for both tracks in recording A6, where waveforms visually overlap except for a few outliers. (C) Templates for both tracks in recording A1 are visually distinct. (D) Templates for both tracks in recording A6 show near-complete overlap and similar amplitude, indicating high template similarity.

classification performance. We tested both SpyKING Circus 2 [4] and MountainSort5 [42] with detection thresholds based on the smaller template (−3.0 for A1 and −2.5 for A5). While both algorithms detected only a fraction of our tracked spikes, they struggled to separate distinct tracks and merge dissimilar spikes into a single unit (Table 3). These outcomes propose that full unsupervised pipelines are not suited for microneurography data and support our presented approach of explicitly separating detection and supervised sorting, which allows for greater control and better sorting results. We applied both sorters with only minimal parameter adjustments to reflect a straightforward use case, acknowledging that further tuning may improve the performance, but the core issue of the lack of separation between fibers remains.

## Discussion

To the best of our knowledge, this work is the first systematic approach analyzing spike sorting of microneurography recordings. It stands out by harmonizing microneurography data from two geographically distant locations and independent work groups with different recording hardware and software [43]. We incorporated 26 datasets, which capture the typical broad spectrum of variability inherent to microneurography recordings. The dataset diversity was carefully curated to include a wide range of recording durations and number of tracks in multi-track recordings, but excluded recordings where spikes overlapped, as these instances could cause even more challenges for sorting. Spikes recorded via microneurography are highly sensitive to different factors, such as electrode movement or environmental electrical noise and spikes from different nerve fibers can be remarkably similar in shape. Our analysis aimed to address these complexities

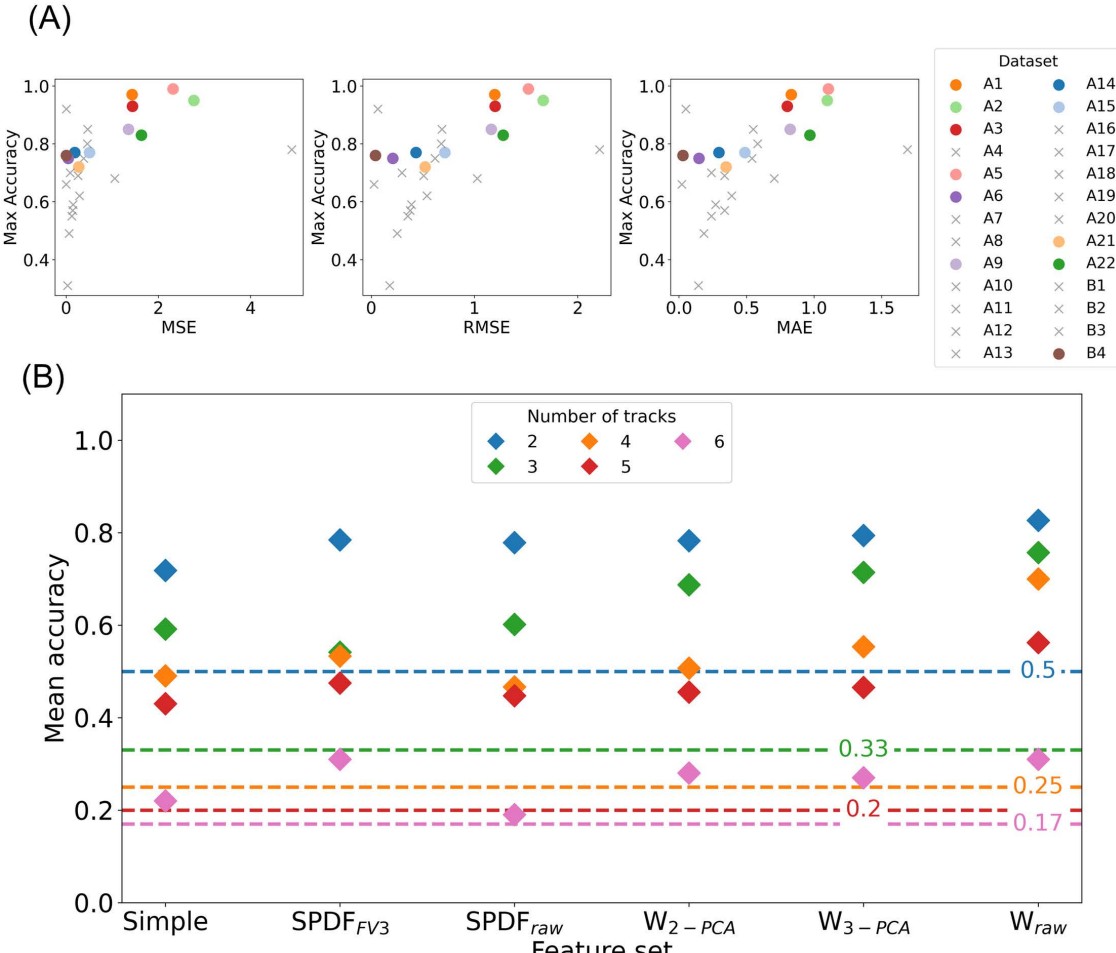

**Fig 4. Indicators of sorting success.** (A) Relationship between template distance and best-achieved sorting accuracy. The distance metrics used are the mean squared error (MSE), the mean absolute error (MAE), and the root mean squared error (RMSE). A smaller distance reflects a higher similarity between the two templates, corresponding to lower sorting accuracy. Smaller distances indicate higher similarity between spike templates and are generally associated with lower classification accuracy. To illustrate this relationship clearly, datasets with only two fibers are shown as primary data points. Additionally, for completeness and transparency, datasets with more than two fibers are included using a workaround: we selected the pair with the highest template similarity and marked these as grey crosses. While this allows all datasets to be visualized, it does not always reflect a fair comparison and may yield contradictory results, as seen in dataset B3, which exhibits very low error scores (e.g., MSE 0.004), but a high maximum classification accuracy (0.92). (B) Mean accuracy of each feature set (x-axis) across recordings, with accuracy values shown on the y-axis. The color of each marker represents the number of fibers tracked in the recording. The markers positioned above the horizontal line indicate that the classifier performs better than the random chance for the corresponding number of fibers.

while providing a transparent and well-documented open-source pipeline, which can be adjusted to other types of neuro-electrophysiological data.

When regarding the mean of all tested recordings, the raw signal feature set ($W_{raw}$) revealed the highest potential for sorting, slightly outperforming the features from the SS-SPDF method. The raw waveform ($W_{raw}$) yielded the best performance in 17 out of 26 datasets when employing SVM classifiers with RBF kernels, with the best accuracy of 0.99 on 2 classes and the worst 0.19 on a recording with 6 classes. However, when comparing the sorting potential of different feature sets for individual recordings, our findings emphasize that the choice of the optimal feature set depends heavily on the specifics of individual recordings, which may exhibit significant variability. A more detailed discussion of related work in the field is provided below.

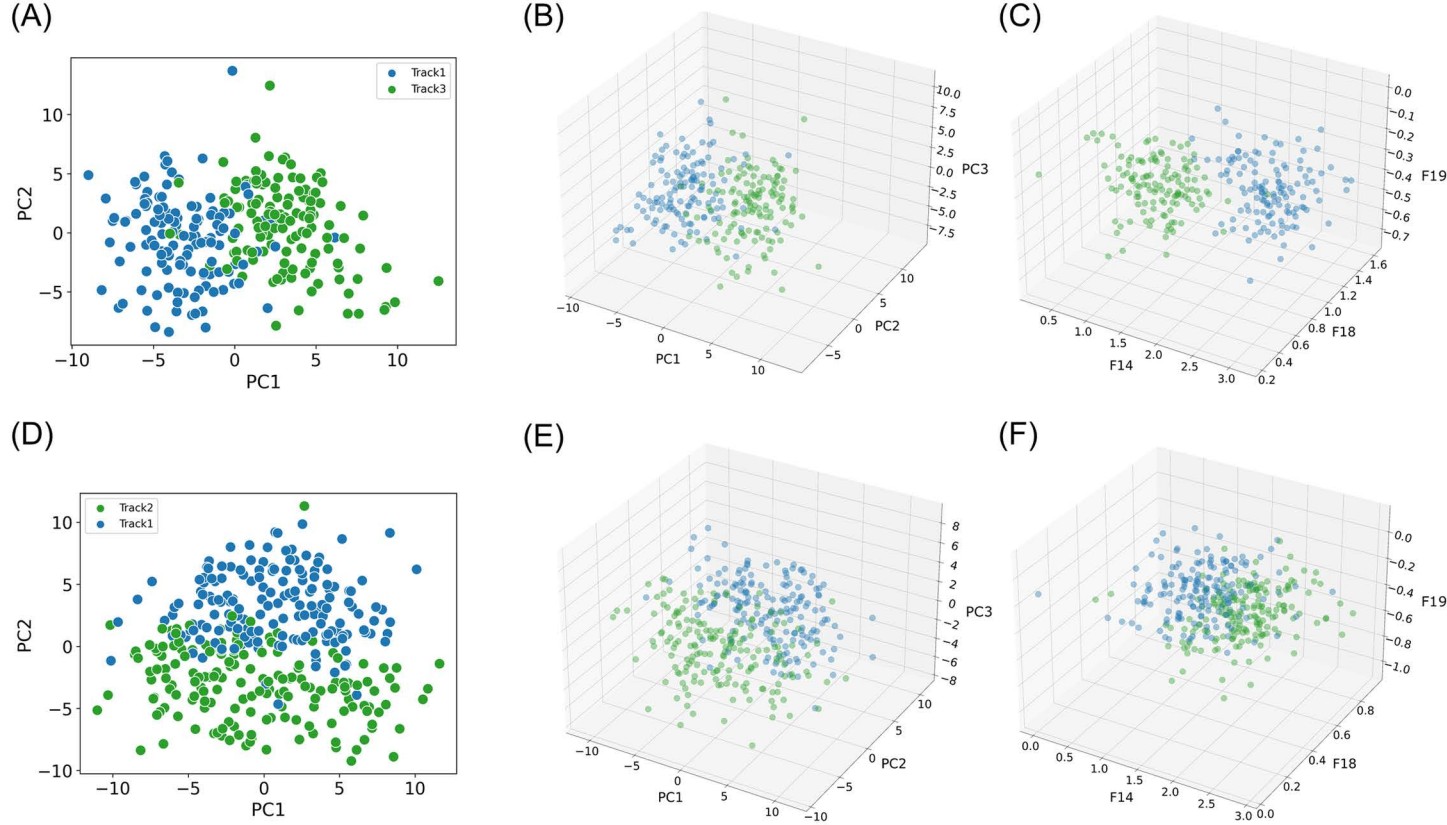

**Fig 5. Comparison of unsupervised clustering challenges for datasets A1 and A3.** Each panel visualizes spikes in different feature set spaces, color labels are applied based on the ground truth labels obtained via the marking method. Panels (A-C) show results from dataset A1. (A) PCA projection in two components (B) PCA projection in three components (C) three-dimensional feature set SPDF$_{FV3}$. Panels (D–F) correspond to dataset A3. (D) PCA projection in two components (E) PCA projection in three components (F) three-dimensional feature set SPDF$_{FV3}$. These visualizations highlight the difficulty of distinguishing clusters using low-dimensional PCA representations, as well as in the SPDF$_{FV3}$ feature space for dataset A3. Clear separation between clusters is observed only in the SPDF$_{FV3}$ representation for dataset A1 (panel C), while the other views show substantial overlap.

**Table 2. Clustering performance across increasing numbers of PCA components for A1.** To assess whether additional components improved separability, PCA was extended up to eight components beyond the commonly used first two or three. Clustering performance was evaluated using Adjusted Rand Index (ARI), Normalized Mutual Information (NMI), and V-measure.

| Number of PCA components | ARI | NMI | V-measure |
|---|---|---|---|
| 2 | 0.70 | 0.59 | 0.59 |
| 3 | 0.70 | 0.59 | 0.59 |
| 4 | 0.71 | 0.61 | 0.61 |
| 5 | 0.73 | 0.62 | 0.62 |
| 6 | 0.73 | 0.62 | 0.62 |
| 7 | 0.73 | 0.62 | 0.62 |
| 8 | 0.73 | 0.62 | 0.62 |

The transparency of the computational pipeline allowed us to trace the relationships between the spike waveform similarities and the resulting sorting difficulties. Our work represents an important first step towards automatized spike sorting in microneurography. We demonstrated that a classifier trained on a subset of tracked spikes achieves promising results

**Table 3. Performance of sorting algorithms within the SpikeInterface framework for datasets A1 and A5.** SpyKING Circus 2 and MountainSort5 were applied to two microneurography recordings (datasets A1 and A5). While both algorithms detected varying numbers of units and identified only a subset of the ground truth spikes, the primary issue was the failure to distinguish between fibers: spikes from both fibers were consistently merged into a single unit.

| Dataset | Sorting algorithm | Number of tracks | Total number of true spikes | Detected units | True positive detected spikes | Correct fiber assignment |
|---------|-------------------|------------------|-----------------------------|----------------|-------------------------------|--------------------------|
| A1 | SpyKING Circus 2 | 2 | 256 | 3 | 46 (Track1: 20, Track2: 26) | No, both fibers merged to single unit |
| | Mountain Sort5 | | | 5 | 12 (Track1: 8, Track2: 12) | No, both fibers merged to single unit |
| A5 | SpyKING Circus 2 | 2 | 340 | 2 | 99 (Track2:50, Track3: 49) | No, both fibers merged to single unit |
| | Mountain Sort5 | | | 4 | 34 (Track2:16, Track3: 18) | No, both fibers merged to single unit |

when applied to other tracked spikes, and hence, has high potential to sort untracked spikes correctly as well. Importantly, the sorting accuracy on tracked spikes will further allow us to estimate the "sortability" of a recorded file providing us with important knowledge on the reliability of sorting unlabeled data (untracked spikes). In the case of microneurography, with the standard presence of baseline electrical stimulation every 4 seconds, we can obtain 80–100 labeled spikes after 320–400 seconds of stimulation, which is typically sufficient for training of low-parameter classifiers (such as SVM) and making our approach practical for many microneurography labs.

In contrast to our approach, applying full spike sorting pipelines, such as those offered through SpikeInterface with multiple state-of-the-art sorters, resulted in poor performance when used on example microneurography data. Even high-quality recordings had low detection rates for our previously tracked spikes and produced unreliable sorting results using SpyKING Circus 2 and MountainSort5, which are based on clustering combined with template matching [4] and ISO-SPLIT clustering [42], respectively. Although the comparison was done on a limited subset of the available data and without extensive parameter tuning, these findings highlight the limitations of unsupervised end-to-end methods for microneurography data, where single recording electrode, low signal-to-noise ratios, and similarity of spike shapes of different nerve fibers complicate fiber differentiation.

Feature set extraction methods showed substantial variation in performance across datasets. Although $W_{raw}$ generally outperformed other feature sets, it was not consistently optimal, underscoring the need for dataset-specific approaches. For comparison, we also evaluated all feature sets using a random forest classifier. Notably, this approach yielded higher accuracy for $SPDF_{raw}$ (mean accuracy 0.64 for random forest, 0.62 for SVM), while the remaining feature sets showed comparable performance to the SVM classifier. This suggests potential advantages of specific classifier-feature set combinations in certain cases. Full performance metrics for the random forest classifier are provided in the Supporting Information (S10–S13 Files). Phase space representation originated from the works of Aksenova et al. [9] and Chivirova et al. [10], and was implemented in our work following the SPDF features introduced by Caro-Martín et al. [11], remains a powerful alternative and also showed good performance in our work. By shifting the time-component into the curve-parametrization parameter role, we consider the phase space representation to be a strong candidate for the development of the computational pipeline trained on multiple recordings. Template matching approaches in phase space should also be tested as computationally efficient and transparent alternatives.

Machine learning, such as automatic feature extraction [44] and multi-task transfer learning [45] could offer an effective strategy for combining multiple recordings into a unified model despite the observed sensitivity to feature set selection. These approaches can make the use of more advanced models increasingly feasible, for example, preliminary tests with Variable Projection Networks (VPNets) [46,47] indicated promising generalization potential in multi-record spike sorting.

An additional point, not addressed in our work, remains the challenge of overlapping spikes. When targeting the analysis of spontaneous activity in microneurography recordings, the issue of overlapping spikes becomes unavoidable and poses a significant challenge for spike sorting, which is also mentioned by Aksenova et al. [9]. In our current setup, this issue is mitigated using evoked activity and different stimulation protocols. Even when two fibers produce spikes with identical latencies, superimposed spikes can still be differentiated due to varying amounts of activity-dependent conduction velocity slowing over time [48]. This effect is clearly visible in the waterfall representation and allows us to distinguish between fibers based on how their conduction velocity changes in response to different types of stimulation.

However, as we move toward the task of sorting spontaneous activity, where stimulation-derived timing information is unavailable, overlapping spikes become an issue. In such cases, relying solely on the time domain representation of spike waveforms may no longer be sufficient. Small distortions in waveform shape due to overlapping spikes or intrinsic variability can lead to misclassification. Incorporating phase space-based features, potentially in combination with fiber conduction velocity information, available in microneurography, could be efficient for extending our supervised spike sorting framework to reliably detect and classify overlapping spikes during spontaneous bursts.

While no direct result comparison to the literature is possible, we would like to link our findings regarding the methodological choices.

Several other works report SVMs as a strong tool to assess classification performance from ground truth or tracked spikes. Fournier et al. take the unsupervised approach by relying on consensus clustering without making statistical assumptions about spike shapes [14]. Their method builds robust clusters through repeated k-means runs and then leverages template matching based on these clusters. Interestingly, they use SVMs to compute an upper bound for sorting accuracy, similar to our use of classifiers on subsets of tracked spikes to estimate performance for future analysis of untracked spikes. However, their reliance on PCA for feature extraction may not be optimal for our data, where alternative features provide better separation.

Vogelstein et al. explore the use of GiniSVMs for both spike detection and classification, using 1.25 ms waveform segments as input features [15]. Their two-stage approach first distinguishes spikes from noise and then classifies the spike waveform into templates. Compared to conventional template matching, the SVM consistently outperforms across varying signal-to-noise ratios, which aligns with our own observations when investigating template-based methods in the time domain. The use of GiniSVM specifically allows for probabilistic outputs, which adds robustness in uncertain conditions and offers a valuable confidence measure during classification. Given these advantages, it could be worthwhile to test GiniSVM in our pipeline, particularly for scenarios requiring confidence-weighted decisions or where we only have labeled data for a subset of spikes.

While our approach relies on supervised learning, unsupervised Bayesian algorithms showed promising results for noisy neural recordings. Takekawa et al. proposed a spike sorting method using wavelet-transformed features and robust variational Bayes (RVB) clustering based on Student's t-mixture models [12]. Takekawa et al. later introduced an enhanced framework combining multimodality-weighted PCA (mPCA) with an explicit variational Bayes implementation [13]. This approach improves feature selection by emphasizing multimodal components and achieves robust clustering even for bursting and sparse-firing neurons, which we would like to test in the future, as it might have the potential for improving spike sorting for bursting activity from spontaneously active nerve fibers.

While our findings provide important insights into spike sorting for microneurography recordings, the main limitations must be acknowledged. First, although numerous spike sorting algorithms and feature extraction methods exist, our work focused on a narrow selection employing a supervised approach using support vector machines based on our prior experience with unsupervised tools, such as SpikeInterface or k-means clustering with PCA-based features.

Second, our analysis was limited to 26 microneurography recordings. Given the high variability inherent in peripheral nerve recordings, both between participants and across experimental sessions, our findings may not generalize across all microneurography recordings. The features and sorting strategies that proved effective in this data collection may not perform equally well in recordings with different noise characteristics or human conditions.

                                                                 

Third, while we incorporated both raw waveform features and phase-based features, it is important to note that the raw waveform features and their PCA components lack direct physiological interpretability. We acknowledge this limitation, and in comparison, SPDF features might be a more suitable choice, particularly for handling spontaneously evoked spikes that may overlap in time.

The main limitation of our study is the challenge of reliable testing on the unlabeled data, for example, untracked spikes of spontaneous fiber activity. In this work, we report the sorting results on tracked spikes, provided by "ground truth" data. Using 5-fold cross-validation with an 80/20 training and testing split allows us to provide a robust and reliable estimate for classifier performance by reducing potential biases of a single train/test division. While this approach helps assess generalization on labeled data, it does not guarantee comparable performance when applied to untracked spikes. The results presented here should therefore be interpreted as proof-of-concept, demonstrating the feasibility of using labeled spikes as a training set to evaluate the potential of a recording for sorting and training of a supervised classifier. In a real-world application, all available labeled spikes would be used for training for the classifier and then applied to sort the remaining unlabeled spikes detected post hoc. Currently, we are working on the creation of another data set, specifically focused on non-tracked spikes. This can be achieved by combining experimental protocols simulating spontaneous activity (for example, chemical stimulation) with manual data labeling.

Another drawback of our approach is that the performance of the SVM classifier significantly decreases as the number of fibers in a recording increases (Fig 4B), which aligns with the documented issue in central nervous system recordings [49]. One potential approach to mitigate this challenge is a one-vs-all strategy, where only spikes from a single nerve fiber with a high signal-to-noise ratio are prioritized for sorting. However, some recordings may be too complex or noisy to allow for effective spike sorting, making them unsortable. It is, therefore, crucial to raise awareness among experimenters about the importance of minimizing the number of fibers in recordings when reliable spike sorting is essential. Prioritizing recordings with fewer fibers can help ensure more accurate and consistent spike sorting results.

## Conclusions and outlook

We conducted the first systematic investigation of spike sorting perspectives and challenges in extracellular single nerve fiber recordings from peripheral nerves. By harmonizing data between two microneurography labs and using an electrical stimulation protocol based on the marking method, we created a diverse ground-truth data collection. We designed an open-source computational pipeline to test various sorting approaches. Due to the high variance between datasets, we used individually trained models. The performance on each dataset largely depended on the number of neural fibers contributing to the single-channel signal and the morphological differences between spikes from different fibers.

Our results show promising performance of the supervised approach to spike sorting in microneurography, with the SVM algorithm applied to the raw spike waveform features slightly outperforming other tested combinations in comparison to other extracted feature sets. Due to the high variance between datasets, we used individually trained models. The performance on each dataset largely depended on the number of neural fibers contributing to the single-channel signal and the morphological differences between spikes from different fibers.

Moving forward, our next goal is to extend the pipeline by incorporating spike detection for untracked spikes. The critical bottleneck in advancing this work is designing experiments where ground truth can be confidently established for all spikes. As a practical solution, we will initially focus on purely electrically evoked activity, where stimulation allows for controlled spike timing. Once validated, we plan to gradually expand to more stimuli, mechanical, thermal, and chemical, that more closely mimic spontaneous activity.

To work toward the goal of sorting untracked spikes, the next step will be to apply our trained classifier to detected spikes that are not aligned on the tracks. While the present study serves as a proof-of-concept, demonstrating the feasibility of using labeled spikes for training, a fully operational pipeline requires robust and reliable spike detection from raw microneurography recordings. This remains a key limiting factor, as accurate detection is especially challenging, as high

noise levels and low signal-to-noise ratios make many spikes small and undetectable using standard detection methods without the aid of the marking method. Addressing this issue is a priority in our ongoing work, as resolving it will be crucial to ensuring accurate comparisons and improving the overall reliability of microneurography data analysis.

In future work, we will explore the performance impact of testing combinations of all feature sets, moving beyond the distinct feature sets analyzed here to gain deeper insights into their contributions to model performance. We plan to benchmark our approach against existing spike sorting frameworks, such as Spike2, in a more extensive and detailed manner, to evaluate their performance and suitability for different aspects of microneurography data, starting from spike detection, which is an important challenge for untracked spikes. This comparison will provide critical insights into standard pipelines' effectiveness and their adaptability to this recording method.

Further, we will explore deep-learning methods and fine-tuning in order to include multi-recording data in one model rather than retraining models for each dataset individually. Additionally, ADS-related parameters, which reflect the number and timing of previous fiber activity [27], could be used to enhance the model's performance via an additional probabilistic layer in the decision process.

The knowledge gained from this work also allows an immediate step towards sorting and analyzing pain and itch-linked peripheral activity, which was not evoked by electrical stimulation. This can be achieved by setting high accuracy thresholds and only analyzing recordings with a satisfactory level of reliability. Furthermore, spike train features should be selected with consideration of their sensitivity to potential misclassifications. For good research practice, the information about the sorting accuracy of the time-locked spiked should be stored together with the spiking activity analysis as one of the assessments of the results' reliability.

Our approach is transferable to other neuro-electrophysiological recordings, particularly when regular stimulation that evokes single spikes can be employed to gain time-locked responses. This enables the generation of a reliable ground truth dataset for method validation. While this study focuses on C-fiber recordings, the same framework could be applied to other peripheral recording types, including Aδ- and Aβ-fibers, as well as sympathetic efferents, broadening its relevance beyond nociception. It could support the development of intelligent prosthetic devices as accurate feedback from the peripheral nervous system is crucial for closed-loop motor control and effective human-machine interaction. For transferring out method to spinal or central recordings, a practicable method for producing time-fixed spikes serving as ground truth data and training set could be developed.

## Materials and methods

### Microneurography

Participants in Aachen were recruited from 01/11/2019–31/03/2023 and participants in Bristol from 04/08/2017–30/11/2022. Twenty-six recordings from healthy volunteers were included in this work. The studies involving human participants were reviewed and approved by the Ethics Board of the University Hospital RWTH Aachen with numbers Vo-Nr. EK141−19 and Vo-Nr. EK143−21 and from the Faculty of Biomedical Sciences Research Ethics Committee at the University of Bristol (reference number: 51882). The participants provided their written informed consent, and the study was conducted according to the Declaration of Helsinki.

In both laboratories, a microelectrode (Frederick-Haer, Bowdoinham, ME, USA) is inserted into the superficial peroneal nerve (Fig 6A) while the volunteer or patient is awake and responsive. In Aachen, the signal is amplified and filtered with a Neuro Amp EX (ADInstruments) amplifier, an additional bandpass filter with 500–1000 Hz, and a 50 Hz notch filter. The recordings were acquired at 10 kHz with an analog-digital converter from National Instruments and a customized software Dapsys (www.dapsys.net) from Brian Turnquist [6,50]. In the laboratory in Bristol, a custom-made recording system and software were used based on electronics and software from Open Ephys [51], APTrack [52], and SpikeSpy [53]. The data was provided in an HDF5 format [54]. This system acquires at 30 kHz, and the signal is digitally filtered (300–6000 Hz). The acquisition board was electrically isolated from the system using a 5kV optoisolator (Intona, Germany). In both

laboratories, the receptive field of C-fibers is determined through transcutaneous electrical stimulation utilizing a Digitimer DS7 constant current stimulator. Once the receptive field is identified, C-fibers are repetitively stimulated at a low frequency (e.g., 0.25 Hz), which enables the marking method to be used. A more detailed description of microneurography can be found here [55].

In this work, spikes were elicited through electrical stimuli. We recorded 22 datasets in the microneurography laboratory in Aachen (A1-A22), and four files were acquired at the lab in Bristol (B1-B4). Details are listed in Table 4. These datasets have different recording conditions, including varying noise levels, signal complexity, and the number of tracks. Each dataset presents distinct challenges, for example, spikes from different tracks have the same amplitude or there are more than two tracks in a single dataset. This diverse collection provided a comprehensive set for evaluating the sorting results.

Table 4 provides a summary of the key characteristics of each dataset, including the number of tracked fibers and the total spike count. The spike count per recording ranges from 206 to 2089, representing the data points that will be split into training and test sets. The number of tracks varies from two up to six, reflecting the complexity of each dataset. The final columns contain the spike numbers for all extracted feature sets.

## Marking method

In microneurography, experimenters employ the marking method [27], a type of stimulation protocol, to observe nerve fiber responses in the form of spikes. The technique utilizes the characteristic that C-fibers have an almost constant conduction velocity in response to repetitive low-frequency stimulation (e.g., 0.125–0.25 Hz), described here as background stimulation. This allows time-locking a subset of spikes. When raw signal segments are plotted sequentially and vertically in a waterfall representation, where each segment starts at the onset of the background stimulus, the spike responses evoked by the background stimulation are vertically aligned (red and green spikes in Fig 6B). This alignment enhances the visibility of spikes, even when the signal-to-noise ratio is poor (Fig 6C for an exemplary spike).

Different C-fibers show distinctive conduction velocities, facilitating the differentiation of multiple fibers within a single recording [6]. When applying further stimulation in the form of extra electrical pulses or natural stimuli, there is a slowing in the conduction velocity of the signal transmission and an increase in latency to the subsequent background stimulus. This phenomenon is known as activity-depending slowing (ADS) [28] and "marks" the fiber. ADS is useful not only to distinguish fibers but also for their classification as different physiological classes of C-fiber exhibit differing degrees of slowing to low and high-frequency electrical stimulation, such as mechanosensitive (CM) or mechanoinsensitive (CMi) C-fibers [55]. In Fig 6B, a representative "waterfall" plot is presented, illustrating the trajectories of two tracked fibers. In this manuscript, we call them *tracks*. When the stimulation remains constant (indicated by blue rectangles), the responses align vertically (lines 1–5). However, after stimulating the fibers with two extra pulses (as seen in line 6), we can observe ADS in both fibers. Through subsequent repetitive low frequency stimulation, latencies recover to their initial values. The remaining issues and challenges in accurately sorting spontaneous firing or chemically induced activity when the spikes have a similar shape persist if in more than one fiber ADS is observed (see line 11). The red and green spikes, representing responses to extra stimuli in line 6, and the orange-marked spontaneous activity in line 11, cannot be reliably sorted as they are not on the tracks.

## Tracking algorithms

When the marking method is applied during the experiments, spike tracks can be efficiently extracted and analyzed post hoc. The tracks are visualized using the waterfall presentation, which makes the nerve fiber responses clearly identifiable. Both laboratories use their respective preferred tracking software and built-in functionality to label spikes along the track. Screenshots of both software can be found in Supporting Information (S15 File).

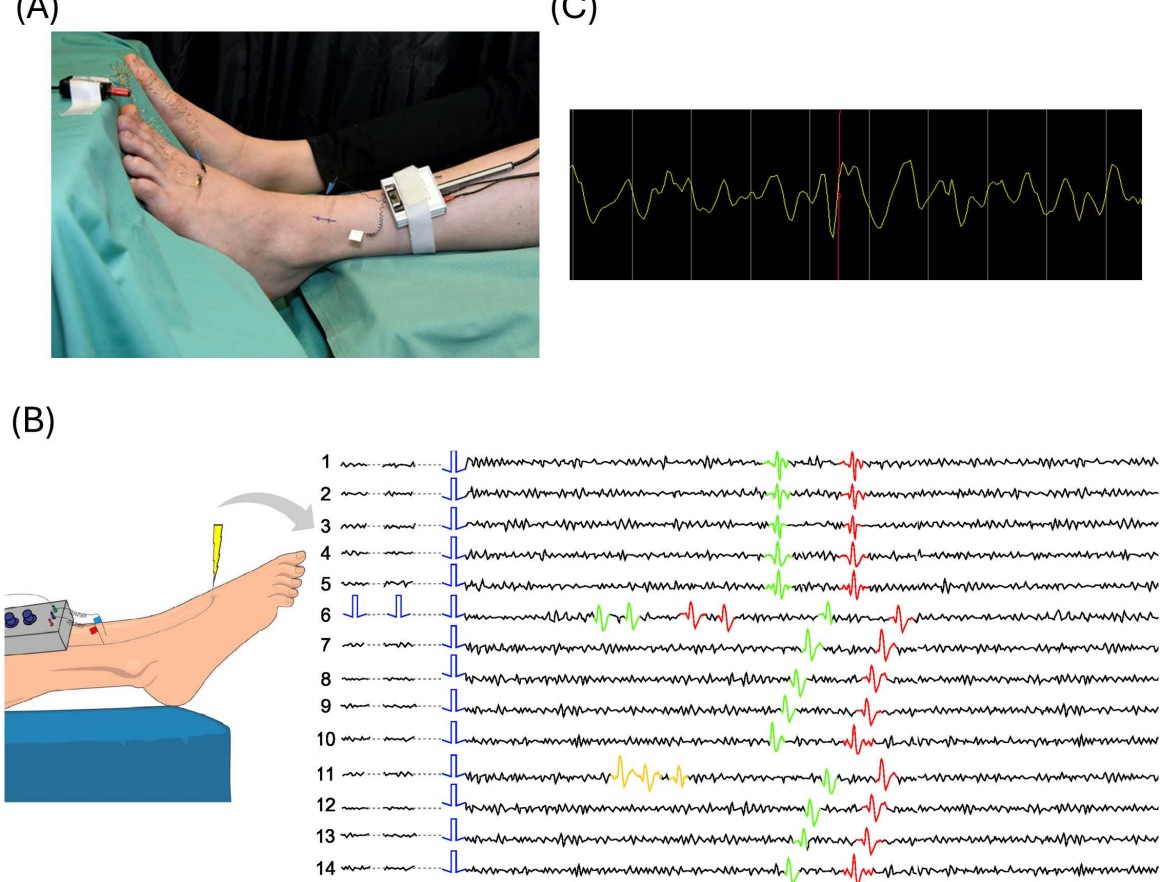

**Fig 6. Details on microneurography experiments.** (A) Set up of microneurography experiments. One microelectrode is inserted into a fascicle of the nerve and another as a reference electrode in the skin nearby. In the receptive field in the skin, C-fibers are activated by electrical stimulation. (B) Schematic waterfall plot of two C-fiber tracks. The waterfall provides a visual representation of the marking method with two active fibers (green and red spikes) represented as track. The onset of the electrical stimulation is indicated by the blue rectangle. Each line, referred to as a trace, begins with the low-frequency stimulus. When extra stimulation is applied (line 6, red and green spikes) or spontaneous activity occurs (line 11, orange spikes), ADS is observed in both fibers [56]. (C) An example spike with a low signal-to-noise ratio. The screenshot of a recording in Dapsys, in which the red line indicates the exemplary spike of interest with an amplitude similar to the background noise level. We could only identify the spike by the marking method.

Although automatic tracking provides a rapid and convenient initial result, it typically requires manual verification and correction to ensure accurate spike identification. Vertical alignment in the presentation facilitates this step, as it indicates the expected location of spikes, even those with low amplitude, making them easier to detect and mark precisely. This manual refinement is critical to ensure each spike has a track label.

In Namer's lab, we used the proprietary software Dapsys, also employed for data acquisition, based on algorithms developed by Brian Turnquist [50]. In Bristol, we developed our own open-source tracking software, SpikeSpy [53], which supports the HDF5/NIX data format. After manual curation, the finalized spike track data can be exported containing the spike time annotations and track labels.

While reliability naturally depends on the researcher performing the curation, the process is generally manageable and tends to involve more routine than complex decision-making.

**Table 4. Overview of the data collection, including the number of active tracks, the class distributions, and the number of total spikes. Additionally, we included the spike numbers for each feature set. The classes are mostly equally distributed. Datasets labeled with A are from the lab in Aachen and datasets labeled with B are from Bristol.**

| Dataset | Number of tracks | Track labels and distribution | Total spikes | Spikes (Simple) | Spikes (SPDF) | Spikes (W) |
|---|---|---|---|---|---|---|
| A1 | 2 | Track1: 129 Track3: 129 | 258 | 258 | 258 | 258 |
| A2 | 2 | Track3: 268 Track4: 268 | 536 | 536 | 536 | 536 |
| A3 | 2 | Track1: 173 Track2: 173 | 346 | 346 | 346 | 346 |
| A4 | 3 | Track1: 647 Track2: 632 Track4: 687 | 1966 | 1916 | 1916 | 1966 |
| A5 | 2 | Track2: 170 Track3: 170 | 340 | 339 | 339 | 340 |
| A6 | 2 | Track2: 103 Track4: 103 | 206 | 206 | 206 | 206 |
| A7 | 3 | Track2: 86 Track4: 86 Track6: 86 | 258 | 258 | 258 | 258 |
| A8 | 3 | Track3: 102 Track4: 97 Track7: 101 | 300 | 298 | 298 | 300 |
| A9 | 2 | Track1: 115 Track2: 115 | 230 | 229 | 229 | 230 |
| A10 | 3 | Track2: 152 Track4: 153 Track5: 114 | 419 | 419 | 419 | 419 |
| A11 | 5 | Track3: 93 Track4: 93 Track13: 93 Track15: 93 Track22: 93 | 465 | 455 | 455 | 465 |
| A12 | 6 | Track3: 348 Track7: 350 Track8: 350 Track11: 349 Track13: 344 Track15: 348 | 2089 | 2056 | 2056 | 2089 |
| A13 | 4 | Track1: 124 Track5: 124 Track11: 129 Track20: 128 | 505 | 499 | 499 | 505 |
| A14 | 2 | Track1: 155 Track2: 155 | 310 | 308 | 308 | 310 |
| A15 | 2 | Track2: 137 Track7: 145 | 282 | 279 | 279 | 282 |
| A16 | 5 | Track1: 159 Track3: 159 Track5: 160 Track7: 157 Track8: 160 | 795 | 785 | 785 | 795 |
| A17 | 5 | Track1: 205 Track2: 152 Track3: 205 Track4: 203 Track6: 198 | 963 | 952 | 952 | 963 |
| A18 | 5 | Track1: 209 Track2: 208 Track3: 209 Track4: 206 Track6: 204 | 1036 | 1020 | 1020 | 1036 |
| A19 | 4 | Track1: 205 Track3: 205 Track4: 204 Track7: 202 | 816 | 809 | 809 | 816 |
| A20 | 3 | Track1: 122 Track2: 112 Track4: 108 | 342 | 337 | 337 | 342 |
| A21 | 2 | Track3: 218 Track4: 308 | 526 | 525 | 525 | 526 |
| A22 | 2 | Track3: 144 Track4: 144 | 288 | 286 | 286 | 288 |

*(Continued)*

**Table 4.** (Continued)

| Dataset | Number of tracks | Track labels and distribution | Total spikes | Spikes (Simple) | Spikes (SPDF) | Spikes (W) |
|---------|------------------|-------------------------------|--------------|-----------------|---------------|------------|
| B1 | 3 | Track1: 277 Track2: 499 Track3: 306 | 1082 | 979 | 979 | 1082 |
| B2 | 5 | Track1: 252 Track2: 254 Track3: 232 Track6: 243 | 981 | 866 | 866 | 981 |
| B3 | 3 | Track1: 189 Track2: 183 Track3: 141 | 513 | 420 | 420 | 513 |
| B4 | 2 | Track1: 184 Track2: 182 | 366 | 332 | 332 | 366 |

### Pre-processing for feature extraction

To ensure data harmonization, we agreed on utilizing the NIX [39] format, a well-established standard in the field of electrophysiology in combination with Neo [57]. A general overview of the preprocessing workflow is shown in Fig 7, while the detailed processing steps are described below.

To handle the datasets generated by Dapsys, we used our Python package PyDapsys [35]. This package enables us free access to electrophysiological recordings stored in Dapsys' proprietary data format. For analyzing an individual recording, we retrieved the raw signal with timestamps and voltages, the timestamps of all tracked spikes with their corresponding track label to ensure ground truth data, as well as the onset timestamps of stimulation events. However, during acquisition, the Dapsys system occasionally introduces short breaks in the recorded signal, resulting in gaps in the raw data. Since Neo [57] expects a continuous time series, these discontinuities cause misalignments between the spike train timestamps and the corresponding signal. To prevent this, we pad the raw signal before writing it to NIX, ensuring a consistent time base. While we can correct the discontinuities after reading the original Dapsys file, the initial step in our workflow still depends on the unmodified raw recording.

Of the four datasets recorded in Bristol, three were provided in HDF5 (.h5) [54] format, which could be easily converted to the NIX format as NIX is based on HDF5. One raw signal recording was available only in MATLAB format, in contrast to the others. To maintain consistency across data handling, this file was read into a data frame and incorporated into the unified NIX structure. However, due to its format and status as a single outlier in the preprocessing workflow, it was excluded from the figure. Additionally, due to differences in the recording setup at Bristol, the voltage polarity of the signals was inverted compared to the Aachen datasets. To correct this, we negated the raw signal before applying a rolling mean filter, smoothing the data for subsequent analysis. The data was processed in pandas data frames containing the raw data, stimulation times, and spike times.

Our next step involved extracting the spike waveforms from the raw signal. As illustrated in Fig 6C, the red line represents the reference point in time, which is typically located near the center of the waveform. Considering an extracellular C-fiber spike width of approximately 3 ms and a sampling frequency of 10,000 Hz for Dapsys files, it is necessary to encompass 30 datapoints from the raw signal to adequately capture the spike. For instance, a spike occurring at timestamp $t$ would lie within the data slice window $[t - 15, t + 15]$. For Bristol files with a sampling frequency of 30,000 Hz, we consider 60 datapoints as suitable, thereby expanding the window range to $[t - 30, t + 30]$.

To ensure the correct alignment of all spikes, we computed the first derivative and aligned the spikes by the maximum negative peak of the first derivative following Caro-Martín et al. [11]. The derivatives are also required for the feature set computation. Furthermore, to enhance the precision of the derivative computations, we resampled the spikes originally recorded with Dapsys from 30 to 60 datapoints with SciPy's [58] resample function.

To gain deeper insights into spike morphology differences obtained via microneurography, as the last step, we computed a "template" by averaging all individual spikes associated with a specific track. This averaging process

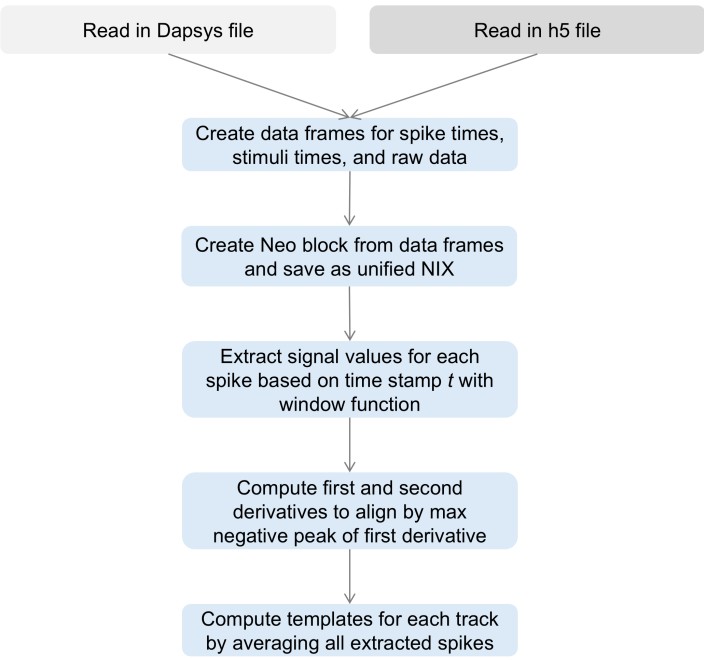

**Fig 7. Details on pre-processing.** The recording (either raw Dapsys file or HDF55 file) is first read in and converted into pandas data frames containing spike times, stimulus times, and raw signals. The data from Dapsys and HDF5 are saved as NIX files via the creation of a Neo block. Spikes are then extracted using a window function around each spike timestamp from the raw signal. The first and second derivatives of the signal are computed to enable alignment based on the most negative peak of the first derivative. Finally, spike templates are computed for each track by averaging all aligned spike waveforms.

resulted in a representative waveform that effectively shows the tracks' distinctiveness or similarity in shape. The templates from two exemplary recordings are shown in Fig 4, while templates from all recordings are provided in the Supporting Information (S5 File). After the pre-processing, we continued with the extraction of feature sets for each spike.

## Feature set extraction

In our analysis, we explored various feature extraction techniques aimed at quantifying the characteristics of spike waveforms (Table 5). We compared the disparities between simpler and more sophisticated feature sets. These extracted feature sets served as inputs for classification methods, and we evaluated their effectiveness in characterizing and discriminating spikes from multiple tracks.

## Amplitude and width – "Simple"

We refer to simple features when taking the most fundamental characteristics of a waveform. Here, we examined two specific features: amplitude $a$ and full-width half maximum (FWHM) $w$. Amplitude is defined as the positive peak voltage value of a spike, while the spike width is defined as the distance between the two nearest points where the signal falls below half of the spike's amplitude. Consequently, each spike can be described by a two-dimensional feature vector. Formally, the feature vector for a spike $S$ is denoted as:

$$S = [a,\ w]$$

**Table 5. Different feature sets are extracted through domain-agnostic dimension reduction as well as through spike approaches, using the raw waveform and computed features on the waveforms. The feature sets include methodologies, such as principal component analysis (PCA) and the features from the spike sorting approach based on shape, phase, and distribution features (SS-SPDF) by Caro-Martín et al. [11].**

| Feature Set | Description | Number of features |
|---|---|---|
| Simple | Amplitude and width | 2 |
| $SPDF_{FV3}$ | Subset of SS-SPDF features (F14, F18, F19) | 3 |
| $SPDF_{raw}$ | Raw SS-SPDF features | 23 |
| $W_{2\text{-}PCA}$ | PCA of raw waveform features (2-comp) | 2 |
| $W_{3\text{-}PCA}$ | PCA of raw waveform features (3-comp) | 3 |
| $W_{raw}$ | Raw waveform features | 30/60 |

## Shape-, phase- and distribution-based features – $SPDF_{FV3}$ and $SPDF_{raw}$

In 2018, Caro-Martín et al. developed a comprehensive spike sorting pipeline for extracellular recordings that outperforms contemporary methods used in neurophysiology [11]. They tested their algorithm on two synthetic datasets as well as on real extracellular recordings of neural activity within the rostral-medial prefrontal cortex from rabbits. Within their methods, they extracted 24 distinct features from each spike on shape-, phase-, and distribution-based characteristics. Shape features describe the waveform of the first derivative in the time domain, while phase features relate the amplitudes of the first and the second derivative to points in the phase space. Additionally, distribution features take into account the amplitude distributions of both the first and second derivatives of each spike.

Caro-Martín et al. employed a modified k-means clustering algorithm that finds the optimal number of clusters and clustering results. This algorithm also addresses the issue of overlapping spikes and evolving waveforms over time. They introduced their own validity score and error index to evaluate their method and compare it with alternative spike sorting methods. Furthermore, the authors emphasize the method's computational efficiency and highlight the enhanced physiological interpretability of their feature-based approach compared to algorithms relying on dimensionality reduction techniques.

In this work, we implemented these features in Python. They represent the more sophisticated approach. In our implementation, we primarily followed the definitions in the paper. However, we had to make certain necessary adjustments.

First, we compute both the first and second derivatives for each spike. Six fundamental points characterize a spike, forming the base for the feature computations. In some of our detected instances, the first fundamental point (the first zero-crossing of the first derivative) was undefined. Consequently, we automatically excluded these spikes from our data. The final numbers are listed in Table 4.

For the final implementation, we had to modify two feature definitions. In the case of Feature 4, it was unclear how the reference waveform was computed. Therefore, we decided to exclude Feature 4 from our final vector.

For Feature 8, the original description referred to it as the "root-mean-square of the amplitudes before the FD event of the action potential" [11]. However, after closer examination of the mathematical formula, it became apparent that the values were not squared, and the authors did not define the variable $m$. We redefined Feature 8 as $f_8 = \sqrt{\frac{\sum_{i=s}^{P1} a_{FD_i}^2}{P1-s}}$, where $s$ is the beginning of the window.

The final feature vector of spike $S$ is denoted as follows:

$$S = [f_1, f_2, f_3, f_5, \ldots, f_{24}]$$

In addition to the full feature vector, Caro-Martín et al. also evaluated a reduced subset of three features (FV3), specifically F14 (positive peak of the spike's first derivative), F18 (positive peak of the second derivative), and F19 (negative

peak of the second derivative), originally proposed by Paraskevopoulou et al. [29,59]. To explore the impact of lower-dimensional input on classification performance, we included this subset as another feature set in our analysis. The final subset feature vector of spike S is denoted as:

$$S = [f_{14}, f_{18}, f_{19}]$$

### Raw waveform feature sets – $W_{raw}$

The next feature set is the raw signal itself, namely $W_{raw}$. We did not process the signal segments further and, instead, directly employed the 30 and 60 voltage values, respectively, as input for each spike waveform compared to [29–31]. As a result, for a given spike denoted as *S,* the feature vector is denoted as follows with *s* describing the voltage at position *x*:

$$S_{Aachen} = [w_1, w_2, \ldots, w_{30}]$$

$$S_{Bristol} = [w_1, w_2, \ldots, w_{60}]$$

### PCA features of raw waveform – $W_{2\text{-PCA}}$ and $W_{3\text{-PCA}}$

Due to the discrepancy in the dimensionalities of the initially presented feature set vectors, we employed principal component analysis (PCA) on the raw waveform. We conducted PCA with both two and three components to reduce the feature vector from its original 30, or 60 dimensions, respectively. As a result, we generated two additional feature vectors for each spike ($W_{2\text{-PCA}}$ and $W_{3\text{-PCA}}$) all of which serve as input for the spike sorting process. The definitions of these feature vectors for the spike are denoted as follows:

$$S_{2\ components} = [PC1,\ PC2]$$

$$S_{3\ components} = [PC1,\ PC2,\ PC3]$$

### Classification

As a classification method, we applied support vector machines (SVMs) motivated by their efficiency and low number of hyperparameters [60]. The models aim to identify a line or hyperplane within the feature space to distinguish between different classes. To assess the accuracy and generalizability of our models, we employed 5-fold cross-validation, which repeatedly partitions the data into training and test sets. We adhered to the default parameters as provided in the sci-kit learn implementation [41].

### Accuracy

To assess our spike sorting results and compare the performance of various feature sets and methodologies, we used evaluation methods tailored to each approach. Accuracy is a metric for evaluating classification results and describes the fraction of correct predictions [61]. We employed accuracy as it is most meaningful when class sizes are relatively balanced. Higher accuracy indicates a better-performing classification model. It is usually expressed as a percentage and is defined as the ratio of correctly predicted instances to the total instances:

$$Accuracy = \frac{Number\ of\ correct\ predictions}{Total\ number\ of\ predictions}$$

(1)

## Statistical analysis

To evaluate whether there were statistically significant differences between the accuracies of the feature sets, we employed the Wilcoxon signed-rank test, a non-parametric test used for comparing two matched samples [62]. This test was selected because it does not assume a normal distribution of the data and is appropriate for comparing paired observations, such as accuracy scores obtained from different feature sets on the same datasets. The statistical analysis was performed with SciPy [41] v1.10.1.

The accuracies for each feature set were compared in a pairwise manner with each pair representing the performance of two feature sets on the same recordings. The test was used to determine whether the distribution of the accuracy differences between the feature sets was statistically significant.

Given that multiple pairwise comparisons were performed, the risk of false positive errors increases. To address this, we applied the Bonferroni correction to adjust the alpha level [63]. Specifically, the initial alpha level ($\alpha = 0.01$) was divided by the number of comparisons to control the error rate ($\alpha_{adj} = 0.01/n$, where $n$ is the number of comparisons). The corrected alpha level was used to determine whether the differences between the feature sets were statistically significant after adjustment for multiple comparisons.

## Measurement of template similarity

To assess the similarity between spike templates, we used several distance metrics to quantify the differences between two spike templates. These templates are the average waveform of spikes associated with individual fibers.

The motivation for using these distance metrics came from the need to investigate whether the similarity between spike templates correlates with the classification accuracy of the feature sets derived from these waveforms. Specifically, we hypothesized that feature sets generated from more similar spike templates would yield lower classification accuracies, while feature sets generated from more distinct templates would lead to higher accuracies.

We compared the distance metrics between spike templates with the best classification accuracies to test this hypothesis. By evaluating the relationship between the template similarities, we aimed to determine whether lower similarity between spike templates predicts better classification performance.

We considered several distance metrics in this analysis, including mean squared error (MSE) [64], mean absolute error (MAE) [65], and root mean squared error (RMSE) [66]. Here, $n$ represents the length of each template. $T = [t_1, \ldots, t_n]$ and $\hat{T} = [\hat{t}_1, \ldots, \hat{t}_n]$ denote the waveform templates being compared.

## Mean squared error

The mean squared error (MSE) is used to quantify the average squared difference between two waveform templates. Smaller MSE values indicate higher similarity and larger values denote greater distinguishability between templates. For this analysis, MSE was selected as it effectively highlights differences in waveform shape.

$$MSE = \frac{1}{n}\sum_{i=1}^{n}(t_i - \hat{t}_i)^2$$

(2)

## Mean absolute error

The mean absolute error (MAE), on the other hand, measures the average absolute difference between templates, offering a straightforward interpretation of the average error without squaring the values.

$$MAE = \frac{\sum_{i=1}^{n}\left|t_i - \hat{t}_i\right|}{n}$$

(3)

## Root mean squared error

The root mean squared error (RMSE) provides a distance measure in the same units as the original data, making it easier to interpret in practical terms.

$$RMSE = \sqrt{\frac{\sum_{i=1}^{n} (t_i - \hat{t}_i)^2}{n}}$$

(4)

## Supporting information

**S1 File. SVM classification accuracy scores.** Accuracy values for SVM classification, provided as a CSV file with comma-separated values.
(CSV)

**S2 File. SVM classification macro-averaged F1-scores.** Macro-averaged F1-scores values for SVM classification, provided as a CSV file with comma-separated values.
(CSV)

**S3 File. SVM classification precision scores.** Precision values for SVM classification, provided as a CSV file with comma-separated values.
(CSV)

**S4 File. SVM classification recall scores.** Recall values for SVM classification, provided as a CSV file with comma-separated values.
(CSV)

**S5 File. Spike templates for all datasets.** The spike templates were computed by averaging all tracked spikes after aligning them to the time point of their maximum negative peak. These templates represent characteristic waveforms for each identified track across recordings and give insights into morphological differences.
(PDF)

**S6 File. Cumulative explained variance ratio.** The cumulative explained variance ratio for principal component counts ranging from 2 to 8, visualized separately for datasets A1 (S6.1) and A3 (S6.2). The plots illustrate how the proportion of total variance captured increases with the number of PCA components, providing insight into the dimensionality required to represent the spike waveform effectively.
(PDF)

**S7 File. K-means clustering ARI scores.** Adjusted Rand Index (ARI) values for k-means clustering, provided as a CSV file with comma-separated values.
(CSV)

**S8 File. K-means clustering NMI scores.** Normalized Mutual Information Score (NMI) values for k-means clustering, provided as a CSV file with comma-separated values.
(CSV)

**S9 File. K-means clustering V-measure scores.** V-measure values for k-means clustering, provided as a CSV file with comma-separated values.
(CSV)

**S10 File. Random Forest classification accuracy scores.** Accuracy values for Random Forest classification, provided as a CSV file with comma-separated values.
(CSV)

**S11 File. Random Forest classification macro-averaged F1-scores.** Macro-averaged F1-scores values for Random Forest classification, provided as a CSV file with comma-separated values.
(CSV)

**S12 File. Random Forest classification precision scores.** Precision values for Random Forest classification, provided as a CSV file with comma-separated values.
(CSV)

**S13 File. Random Forest classification recall scores.** Recall values for Random Forest classification, provided as a CSV file with comma-separated values.
(CSV)

**S14 File. Error values with max accuracy.** Computed error metrics for all datasets, along with the maximum achieved accuracy for SVM classification. These metrics were used to investigate how template similarity affects sorting quality.
(CSV)

**S15 File. Screenshots of Dapsys and SpikeSpy.** To illustrate the tracking process, this file includes two screenshots, one from Dapsys and one from SpikeSpy. Both software tools implement similar tracking mechanisms to extract vertically aligned spike waveforms during microneurography recordings, providing experimental ground truth for subsequent analysis.
(PDF)

**S16 ZIP Folder. Feature set data for datasets A1, A3, and A6.** This archive includes the raw extracted feature sets for A1, A3, and A6, as well as $SPDF_{FV3}$ features for A1 and A3. It also contains waveform data used for plotting and template computation in Fig 3, along with the data used for PCA and clustering analyses presented in Fig 5.
(ZIP)

## Acknowledgments

We acknowledge Aidan Nickerson and Danxia Bao for their technical support. We would also like to acknowledge the three Reviewers for providing feedback on many aspects of our work. Thanks to this feedback, we were able to improve the quality and reproducibility of our code, as well as improve the methodology of our spike alignment.

## Author contributions

**Conceptualization:** Ekaterina Kutafina, Barbara Namer.

**Data curation:** Andrea Fiebig, James Dunham, Barbara Namer.

**Formal analysis:** Alina Troglio, Peter Konradi, Ariadna Pérez Garriga, Ekaterina Kutafina, Barbara Namer.

**Software:** Alina Troglio, Peter Konradi, Ariadna Pérez Garriga.

**Supervision:** Rainer Röhrig, James Dunham, Ekaterina Kutafina, Barbara Namer.

**Visualization:** Alina Troglio.

**Writing – original draft:** Alina Troglio.

**Writing – review & editing:** Alina Troglio, Peter Konradi, Andrea Fiebig, Ariadna Pérez Garriga, Rainer Röhrig, James Dunham, Ekaterina Kutafina, Barbara Namer.

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
