## [Decision Letter · Decision Letter 0]

3 Apr 2025

PONE-D-25-04960Supervised Spike Sorting Feasibility of Noisy Single-Electrode Extracellular Recordings: Systematic Study of Human C-Nociceptors recorded via MicroneurographyPLOS ONE

Dear Dr. Troglio,

Thank you for submitting your manuscript to PLOS ONE. After careful consideration, we feel that it has merit but does not fully meet PLOS ONE’s publication criteria as it currently stands. Therefore, we invite you to submit a revised version of the manuscript that addresses the points raised by the reviewers.

 Please submit your revised manuscript by May 18 2025 11:59PM. If you will need more time than this to complete your revisions, please reply to this message or contact the journal office at plosone@plos.org . Please include the following items when submitting your revised manuscript:

We look forward to receiving your revised manuscript.

Kind regards,

Nicholas V Swindale

Academic Editor

PLOS ONE

Journal Requirements:

2.Thank you for stating the following in the Competing Interests section: [BN received consulting fees from Vertex. The other authors declare no competing interests.]

3. We noted in your submission details that a portion of your manuscript may have been presented or published elsewhere. [Figure 6B has been adapted from our previously published manuscript under a Creative Commons license. The original work has been properly cited and referenced in the manuscript, ensuring compliance with publication guidelines.] Please clarify whether this [conference proceeding or publication] was peer-reviewed and formally published. If this work was previously peer-reviewed and published, in the cover letter please provide the reason that this work does not constitute dual publication and should be included in the current manuscript.

Reviewers' comments:

Reviewer's Responses to Questions

**Comments to the Author**

1. Is the manuscript technically sound, and do the data support the conclusions?

Reviewer #1: Yes

Reviewer #2: Partly

Reviewer #3: Partly

2. Has the statistical analysis been performed appropriately and rigorously? 

Reviewer #1: Yes

Reviewer #2: Yes

Reviewer #3: Yes

3. Have the authors made all data underlying the findings in their manuscript fully available?

Reviewer #1: No

Reviewer #2: Yes

Reviewer #3: Yes

4. Is the manuscript presented in an intelligible fashion and written in standard English?

Reviewer #1: Yes

Reviewer #2: Yes

Reviewer #3: Yes

5. Review Comments to the Author

Reviewer #1: The manuscript presents a supervised pipeline for spike sorting targeting recordings with microneurography that employ a marking method.

In order to overcome the challenges related to extracellular recordings from microneurography, such as very low SNR and spatial resolution (single electrode), the authors suggest a supervised approach that uses the spikes from pre-identified units with the marking method as ground truth to build an SVM classifier that can be applied to spontaneous spikes. Several feature extraction methods are compared, showing a general better performance using the entire waveform data. However, given the substantial inter-subject/experiment variability, the authors also suggest that the proposed method could be used to assess the overall "sortability" and to find the best set of features of a specific recording.

I found the manuscript well structured and well written and I have some a few points and suggestions.

1. When discussing the limitations of the PCA methods in Figure 5, the authors claim that dataset A1 would not yield good results with an unsupervised approach because the clusters overlap. I think that this would be strobgly dependent on i) the number of PC used, and ii) the explained variance of the chosen number of PC. Could the authors show how the clusters look using 3 PCs? What about more PCs? Would a clustering method correctly find the two units if, for example, 5 dimensions were used?

In the same figure, I found panels C-D quite puzzling. The blue and green sub-clusters that the authors state are from the same unit will likely have very different waveforms/templates? How do the authors explain this? Is this something commonly observed in spikes from C-Nociceptors? Could it be that two different fibers are elicited from the same stimulus and have the same latency?

2. In cases where the waveforms/templates are very similar (e.g., Figure 3B-D), is it possible that an individual stimulus results in a fiber firing twice, in sequence? With the marking methods, these two spikes would be classified as separate units, and as the authors suggest, it would be virtually impossible to spike sort them accurately. On the other hand, isn't it unlikely, given the geometry of the fibers and the relative position to the electrode, that the spikes come indeed from the same fiber?

3. Figure 4A. the authors restrict this similarity analysis to datasets with 2 tracks. I think that the number of data points could be extended also to multi-track dataset by considering, for each unit, a second unit with the lowest similaroty (worst case scenario).

4. Since the authors suggest that the method could be applied to spontaneous spikes as well, it would be nice to see that in action. How many spontaneous spikes could be classified on the used datasets?

Minor points:

1. L142: SS-SPDF: first time this acronym is used. May be worth spelling it out

2. L352: why do the authors state that the study is about "spike morphology"? I would classify it as spike sorting analysis only

3. L376-378: what other applications could it be transfered to?

4. Table2: I would rename the Track*, R/S/T, Unit* consistently. The different naming makes it more confusing. You could just use Track1/2/3 or Unit1/2/3

5. L481: Panda's -> pandas

6. Table 3: since most of steps are the same, a diagram/flowchart would be easier to understand. The differences are minor and could be explained in the text/captipn

7. L518-519: this width is usually referred to as FWHM (full-width half maximum)

8. L558: What is W2? I guess that Wraw is the entire waveform (30-90 points), but W2 is not specified

9. L569: where does the 23 come from? Aren't the new features 2 (S2/S3) instead of 4?

Reviewer #2: In this manuscript, the authors describe and evaluate the feasibility of a novel approach to sort spikes in electrophysiological recordings from nociceptive C-fibers in the peripheral nervous system, obtained using microneurography. The approach involves using SVM for the supervised classification of previously labeled spike data, opposed to existing unsupervised methods. The labelling of spikes is performed by employing the marking method, which leverages the difference in conduction velocities between fibers recorded simultaneously by an electrode and the activity-dependent slowing phenomenon in C-fibers. In the marking method, action potentials belonging to a single C-fiber are time-locked to a background low-frequency stimulus. By visualizing the recordings after several stimulation pulses in parallel, spikes originating from the same fiber will apear aligned in time (named a “track”), allowing to assign each spike to an individual fiber, and to extract the spike waveforms from the raw data. From the spike waveform data, the authors extract distinct feature sets (simple features, SS-SPDF features, and raw waveform features), which are the inputs to the SVM classifier. The SVM training/testing is performed for each recorded dataset (26 in total, from two different labs) using 5-fold cross validation, allowing to evaluate the sorting peformance of each feature set on a per-dataset basis, which could be used to guide the selection of the best features for sorting in a given dataset. In the end, the authors analyze and discuss the classification performance of each feature set according to general data features, such as number of individual fibers recorded and similarity of spike waveforms across the fibers in a single recording. They also highglight how the supervised method can be more efficient to sort spikes from different fibers compared to methods based on PCA followed by unsupervised clustering.

The manuscript is relevant for research involving microneurography and may advance the analysis of data from recordings of C-fibers. Moreover, the authors aimed to make the software code to apply the proposed method available, and describe efforts to make the analysis pipeline usable with different proprietary input file formats (leveraging open standards for data and metadata such as odML and NIX). However, there are points that need improvement and clarification in the manuscript.

1. The whole analysis rely on labeled data for ground truth using the marking method, but the manuscript does not detail how the spike tracks were actually obtained. Although the authors took their time to explain the rationale of the method, with helpful illustrative figures, the section in the methods (lines 448-472) does not detail how the actual procedure is executed with the raw data obtained in a recording from Aachen or Bristol. In the main text, the authors just describe this as a pre-analysis (line 165). In addition, there is no code in the GitHub repository for obtaining that track information (assumed to be stored in a NIX file after initial curation; Table 3). Therefore, the manuscript must present a more detailed description on how the marking method is performed as one part of the whole pipeline to obtain the starting spike data for the classification (e.g., software tools, manual curation steps, detailed signal preprocessing steps and parameters). The code should also be shared or properly referred (if in another repository) to allow full reproducibility of the proposed pipeline.

2. Considering the SVM approach, it is also not clear how much labeling as pre-analysis (line 165) is needed assuming a new analysis is performed. In the results presented, each CV-fold uses 80% of the labeled data for training, and the remaining 20% for testing. In a practical scenario, this would imply that, for an analysis starting from scratch, one would need to label 80% of the spikes (by using the marking method) before trying to classify the others. As the exact procedures are not described, it is very difficult to understand how laborious this pre-analysis is. The authors could also explore how the SVM classifier performance varies using different train/test split ratios. Finally, what is considered “sufficient amount of labeled data per recording” (line 312)?

3. Although this work is also introducing an open-access pipeline to preprocess and standardize the data and metadata for the sorting method, the actual code in the https://github.com/Digital-C-Fiber/SpikeSortingPipeline repository consists only of Jupyter notebooks that apparently do not contain all the steps described in the Methods (Table 3). There is only a notebook for one of the proprietary formats (Dapsys), and apparently there is no code for the generation of the NIX files.

4. The use of the provided code is not straightforward. The README file describes only how to create the Python environment, and not how to setup the analysis using an input file in Dapsys or OpenEphys formats. Although the notebook provided for Dapsys provides extensive text descriptions throughout the steps, the cells are composed of many function declarations embedded within the calls to the actual pipeline steps. It is not clear how to adjust or set parameters. Therefore, the code could be reorganized using more robust approaches for such analysis pipelines (see, for example, Cobrawap: https://cobrawap.readthedocs.io).

5. Why is the Dapsys file still needed to run the pipeline after creating the NIX file (Table 3)? The NIX format allows storing continuous time series and would be more convenient to have a common structure that would serve analyzing data from both recording formats (see, again, the approach in Cobrawap). In Table 3, from line 4 the steps are overlapping, with only a few changes of parameters depending on the sampling frequency of the data. Those could be easily selectable as pipeline parameters passed to the code.

6. Although authors mention in their data availability statement that the patient data in the proprietary formats cannot be shared because they might contain sensitive information, the manuscript suggests that those primary data are converted to the NIX format at the beginning (Table 3). One could design the conversion step to remove any sensitive information. Would it be possible to share the source data as anonymized NIX files? If not, this aspect and reasons should also be clarified in the availability statement. The shared data in the supplement contains only the performance metrics of the trained classifiers, which allows reproducing the presented figures, but not the actual analyses.

7. There are contradictions in the presented results. In Figure 4A, the authors state that only datasets with 2 detected tracks are shown. The figure shows only 10 datasets from Aachen, but the data description in Table 2 shows that dataset B1 from Bristol also has 2 tracks only. In addition, Table 2 presents only the total number of spikes in a dataset, while examining the tables in the supplementary files, it is clear that many were excluded for the actual analysis. Table 2 should also state the final number of spikes (total and per track), and the Methods must describe the preprocessing steps and reasoning to exclude spikes before the analysis.

8. Authors explored distinct feature sets (from simple features such as amplitude and width to the complex parameters of the SS-SPDF method), and the classification using the raw waveform features provided the best classification in the majority of the datasets (as assessed in the statistical evaluation in Table 1). In some datasets, the performance of other feature sets could be marginally better (Figure 2). With that, the selection of the feature set for a given dataset could be tailored to provide the best sorting performance. Moreover, in their discussion, the authors suggest that the accuracy could be used to assess the “sortability” of a dataset (lines 302-306), and that the approach could “significantly enhance the realiability in sorting spikes that are not time-locked to their response latency” (lines 319-321). It is difficult to assess those practical implications beyond this validation study. Moreover, in those speculative statements on how to use the approach to improve sorting, the authors mainly state that the supervised method was better than PCA + unsupervised clustering. Assuming that performing the SVM classification and obtaining a dataset-specific accuracy score requires pre-sorting all the spikes in all tracks, why would sorting with the SVM still be needed? If the results are supposed to illustrate that, once trained, the classifier can correctly identify the spikes from different fibers in non-tracked spikes, the result showing that the “sortability” performance decreases as the similarity of the templates increases (Figure 4A) would also not solve the primary problem of sorting spikes outside the tracks (lines 136-139). First, non-tracked spikes might still be difficult to discern if the waveform shapes are too similar. Second, the spike waveforms can often overlap in spontaneous firing (e.g., line 11 in Figure 6B), and the classifier would still fail. Therefore, it is important that the authors detail their reasoning when discussing the implication of their results outside the scope of the study.

9. One of the main arguments in the manuscript is how the supervised method is more adequate to identify the spikes from each track compared to dimensionality reduction with PCA followed by clustering (Figure 5). However, this “traditional” approach was largely improved over the years, and this may not be a fair comparison. An algorithm that also involves template matching would be closer to the proposed approach and might perform as well in cases where the spike waveforms are different, and could perform well in case of spontaneous activity with overlapping spikes (e.g., line 11 in Figure 6B). Therefore, the strength of the findings would be increased if authors applied some state-of-the art sorters that aimed to improve clustering and the precision of the detection in unsupervised settings (e.g., Kilosort 3 and 4, SpyKING Circus, MountainSort). As data is structured in the NIX format, it would be straightforward to use SpikeInterface to apply those sorters in parallel. As the ground truth data is available from the marking method, this would also be descriptive of any limitation of existing sorters in microneurography data.

10. Have the authors consider investigating spike sorters that leverage the temporal alignment of spikes in multichannel recordings? The “waterfall approach” resembles recordings with n-trodes or high-density probes, where multiple electrodes capture spikes originating from a single source. The sorter algorithm takes the electrode distribution and the temporal alignment of spikes into consideration. Therefore, one could speculate that epochs 1-5 in Figure 6 could be structured as if they were obtained by a 5-trode probe and such sorting might be as good as peforming the supervised classification, assuming waveforms are different. This would identify the different templates shown in either green or red in the traces in Figure 6B. Once those templates are known, the templates could be used to match any other spike in the recording using template matching.

11. The goal of spike sorting is to obtain the assignment of each spike that appears in the raw signal to the relevant unknown unit (or fiber, represented by a track in this study). However, with this perspective, it is unclear how the proposed method can be effectively used for spike sorting in a new dataset and how much of improvement it brings to the field. Overall, the manuscript can be confusing regarding the specific goals of the study regarding spike sorting. For example, in the introduction, the authors mention that a primary challenge is to sort the untracked spikes (lines 136-139), and also highlight this when describing the marking method (lines 468-472). However, there is no demonstration on how the method solves this problem. One would expect that the supervised classification method would then help identifying spikes arising from spontaneous activity (e.g., the yellow waveform in Figure 6B, line 11). But all the analyses presented involve tracked spikes only. From the title and abstract, and considering the results, it seems that the main intention of this study is to evaluate and validate the classification of spike data assuming that they can be “tracked” (i.e., a very specific subset of data in a microneurography recording executed together with a specific stimulation protocol). In this case, the text should be revised so that it is more assertive towards that goal, and the implications/challenges that are known but still not addressed are described and discussed in detail only in the Discussion section. And in the discussion, the authors need to be less shallow, and effectively provide insights on how their classifier would be helpful to improve spike sorting in microneurography.

Reviewer #3: COMMENTS TO THE AUTHORS:

Comments to the authors of the manuscript entitled "Supervised Spike Sorting Feasibility of Noisy Single-Electrode Extracellular Recordings: Systematic Study of Human C-Nociceptors recorded via Microneurography" by Alina Troglio, Peter Konradi, Andrea Fiebig, Ariadna Pérez Garriga, Rainer Röhrig, James Dunham, Ekaterina Kutafina, Barbara Namer, to be considered for publication in PLoS ONE (ref. PONE-D-25-04960).

GENERAL COMMENTS:

This study is of relevance for the neuroscience community, especially for researchers interested in improving the spike sorting performance for raw microneurography data (i.e., human multi-unit electrophysiological extracellular nerve recordings). In my opinion, the proposed approach could be interesting for many clinical neurophysiologists for its practical uses for studying peripherical sensory systems beyond the mere spike classification approaches offered by other sophisticated and often opaque spike-sorting methods/algorithms.

According to what the authors stated, the main contribution of this study is the adaptation of lightweight feature extraction methods to individual C-nociceptor microneurography recordings and assessing the classification feasibility based on achieved accuracy and the research question before proceeding with sorting of non-time-locked spikes. The proposed approach is interesting, the manuscript is well written and appropriately illustrated and the authors conducted an acceptable literature review. However, despite what has been highlighted above, I have major concerns with the manuscript in its current version and there are some key aspects that need to be clarified by the authors. Therefore, this paper would deserve publication in PLoS ONE journal once the following points are addressed.

MAJOR POINTS:

P1. First of all, I’m not quite sure how the microneurography recordings are pre-processed and the spikes are aligned. In the sections “Pre-processing for feature extraction”, authors should clarify all the details because the precision in all steps of the spike-sorting procedure critically affects the accuracy of all subsequent analyses.

REGARDING THE ALIGNMENT OF THE SPIKES: when the authors are saying (lines 490-494) “As each spike is identified by a timestamp, we extracted the corresponding index, which we consider to be the center of the spike in the raw signal. This index was then used to determine the voltage values of the spike with the 3 ms window. To ensure the correct alignment of all spikes, we established that this center point index is two positions before the voltage maximum of the spike, which should occur in the rising phase of the spike”; what does this means? Note that, in the SS-SPDF method proposed by Caro-Martín et al. (2018) all extracted spike events (spike first-derivative) were aligned (see Fig. 2d, Left) based on their negative peak positions —a step that improves the classification process. In this way, the phase-space portraits of all the aligned spikes were reconstructed and then twenty-four physiological features (Tables 2 and 3 in Caro-Martín et al., 2018) were extracted from each spike for further spike-sorting processing. If the authors of this manuscript have not correctly aligned the spikes in the first-derivative space with respect to the maximum negative peak, then they have failed in the extraction of all features from SPDF method and, therefore, they have not been successful in all subsequent steps of the spike-sorting analysis.

REGARDING THE FEATURE EXTRACTION: I have serious doubts about the proper extraction of the raw SS-SPDF features. Note that, from a mathematical point of view, the SPDF features proposed by Caro-Martín et al. (2018), or any optimal subset of them, are principal components in an orthogonal space of representation. The independent features (F1–F24) proposed in that article ensure that the feature vector in a 24D-space (R24) does not hold redundant information and thereby remove the need to further reduce the dimensionality following the standard way of the principal component analyses. All the above makes senseless the dimensionality reduction proposed by the authors in this manuscript for the SPDF set of features (what they call SPDF2 and SPDF3). The authors can force a dimensionality reduction but that is redundant and worthless from a physiological point of view. Another crucial aspect that indicates a significant failure in the execution of the feature extraction procedure is the selection of the feature set Wraw (raw waveform features with 30/90 voltage values). The authors say (lines 559-564) “The next feature set is the raw signal itself. We did not process the signal segments further and, instead, directly employ the 30 and 90 voltage values, respectively, as input for each spike waveform. As a result, for a given spike denoted as S, the feature vector is denoted as follows with s describing the voltage at position x”. This proposal is not at all original and directly violates the biophysical and physiological foundations of the generation of action potentials with waveforms discretized in time, whether neuronal or nerve fibers. The key question here is why don't the rest of the authors who systematically work on spike-sorting method/algorithm do this same thing for conforming the spike feature vector? Selecting a feature vector made up of only the sample values of the spike amplitude is substantially incorrect for several reasons: (1) Wraw feature matrix presupposes that all spikes have plausibility in their absolute refractory periods, i.e., the interval between the beginning of the depolarization phase and the end of the hyperpolarization phase. (2) Wraw feature matrix completely ignores the temporal relationships that characterize neural events (with phases of depolarization, repolarization and hyperpolarization that determine a quasi-closed trajectory in the phase-space of a spike) and that differentiate them from other parasitic (non-physiological) waveforms that also make up the recorded spike. (3) Wraw feature matrix assigns the same weight (always equal to 1) to all its features (whether 30 or 90 valtage values) a deficiency that does not allow to evaluate the trade-off between the optimal number of features and the optimal number of electrodes in an array. (4) Finally, it should be added that the alignment of the spikes with respect to the negative peak of the first derivative of the spike also guarantees alignment with respect to the zero-crossing of the second derivative of the spike, a procedure followed in the SS-SPDF method that substantially improves spike clustering and classification, but that was not applied properly in this manuscript.

P2. The spike sorting workflow described here (block diagram in Fig. 1) will suffer from the same limitations as all workflows based on clustering/matching templates techniques in the time domain of the neuronal spikes: the overlapping spikes, in spikes, will be discarded. Because mixtures of overlapping waveforms will be an outlier in the clustering space, they will be discarded, and this is problematic. This is why recent spike sorting algorithms [Caro-Martín et al., 2018] are going into the template-matching directions but in the phase space of the neuronal spikes (see also these key articles [Aksenova et al., 2003; Chivirova et al. 2005] for review). Most of the template-matching based algorithms [Bankman et al. (1993); Lefebvre et al. (2016); Yger et al. (2016); Pachitariu et al. (2016)] construct the templates, compare segments of the signal with all available templates in time domain and then selecting the best matching template. The main drawback of these template matching algorithms on the spike time-domain is that spike waveforms could be slightly distorted not only in amplitude, but also along the time axis. Consequently, classes of spikes may not form clusters in the feature space related to time domain and the distributions inside the classes may not be Gaussian. Supervision by a human expert is required for a correct classification. I would really encourage the authors to include in the Introduction and Discussion sections, analytical and interpretative comments on these three studies [Aksenova et al. (2003); Chivirova et al. (2005); Caro-Martín et al. (2018)] that used Template Matching algorithm, K-means clustering and/or appropriate combinations of them (K-TOPS clustering algorithm) for spike sorting in phase space, for sorting both single-unit spikes and overlapping waveforms. In addition, these authors [Aksenova et al. (2003); Chivirova et al. (2005); and also Caro-Martín et al. (2018)] argue that making template matching on the spike phase-space is more efficient (in terms of execution time and computational complexity) than doing it on spike time-domain. The authors of this manuscript could discuss the possible advantages/disadvantages of implementing template matching method in the spike phase-space on this mixed approach of supervised spike-sorting with machine-learning models for C-nociceptor microneurography data for sorting both single-unit spikes and overlapping waveforms. For this it is essential to consider that the features vector for each spike should also be based on latencies (not solely in 30/90 spike amplitudes) that are the other main features of the spike waveform.

P3. The comparison with other approaches for spike sorting such as Template Matching in spike phase-space [Aksenova et al. (2003); Chivirova et al. (2005], Neural Networks [Werner et al. (2016); Hermle et al. (2005); Kim & Kim (2000)], Support Vector Machines [SVM; Fournier et al. (2016); Jacob-Vogelstein et al. (2004)] or Bayesian Algorithms [Lewicki (1998); Takekawa et al. (2010); Takekawa et al. (2012)] is not done or is done very lightly without going into details. I would advise the authors to elaborate a bit more on the discussion about the practical and methodological advantages/differences of their supervised spike sorting for C-nociceptor microneurography approach with respect to those.

NEW RECOMMENDED REFERENCES:

Aksenova TI, et al. An unsupervised automatic method for sorting neuronal spike waveforms in awake and freely moving animals. Methods, 2003; 30:178–187.

Chibirova OK, et al. Unsupervised Spike Sorting of extracellular electrophysiological recording in subthalamic nucleus of Parkinsonian patients. Biosystems 2005; 79:159–171.

Werner T, et al. Spiking Neural Networks Based on OxRAM Synapses for Real-Time Unsupervised Spike Sorting. Front. Neurosci. 2016; 10:474.

Hermle T, et al. ANN-based system for sorting spike waveforms employing refractory periods.In ICANN 2005 LNCS (Eds Duch, W., Kacprzyk, J., Oja, E. & Zadrożny, S.) 3696:121–126 (Springer, Berlin, Heidelberg, 2005).

Fournier J, et al. Consensus-Based Sorting of Neuronal Spike Waveforms. PLoS One 2016; 11:e0160494.

Jacob-Vogelstein R, et al. Spike sorting with support vector machines. Conf. Proc. IEEE Eng. Med. Biol. Soc. 2004; 1:546–549.

Takekawa T, et al. Accurate spike sorting for multi-unit recordings. Eur. J. Neurosci. 2010; 31:263–272.

Takekawa T, et al. Spike sorting of heterogeneous neuron types by multimodality-weighted PCA and explicit robust variational Bayes. Front. Neuroinform. 2012; 6:5.

MINOR POINTS:

(i) When the authors are saying (lines 37-38) “Our approach provides the foundation for further development of spike sorting algorithms in noisy extracellular recordings of neural activity”; What does this means? I don’t see how this approach provides the foundation for further development of spike sorting algorithms.

(ii) In Fig 4A, part of the legend is cut off, please fix this.

Personally, I encourage the authors to adequately address all these concerns in the manuscript, so that it includes all the necessary details, because I consider this type of mixed approach that integrating supervised spike-sorting with machine-learning models for C-nociceptor microneurography data very interesting and this is irrefutably necessary in clinical neurophysiology.

6. PLOS authors have the option to publish the peer review history of their article (what does this mean? ). If published, this will include your full peer review and any attached files.

**Do you want your identity to be public for this peer review?** For information about this choice, including consent withdrawal, please see our Privacy Policy .

Reviewer #1: No

Reviewer #2: No

Reviewer #3: No

---

## [Author Response · Author response to Decision Letter 1]

30 May 2025

Reviewer #1: The manuscript presents a supervised pipeline for spike sorting targeting recordings with microneurography that employ a marking method.

In order to overcome the challenges related to extracellular recordings from microneurography, such as very low SNR and spatial resolution (single electrode), the authors suggest a supervised approach that uses the spikes from pre-identified units with the marking method as ground truth to build an SVM classifier that can be applied to spontaneous spikes. Several feature extraction methods are compared, showing a general better performance using the entire waveform data. However, given the substantial inter-subject/experiment variability, the authors also suggest that the proposed method could be used to assess the overall "sortability" and to find the best set of features of a specific recording.

I found the manuscript well structured and well written and I have some a few points and suggestions.

We would like to thank the reviewer for their valuable and positive feedback, as well as their insightful suggestions, which helped to improve the quality of our manuscript. Particularly, spotting the false separation within the same cluster on Figure 5 allowed us to identify a computational bug, which we previously overlooked.

1. When discussing the limitations of the PCA methods in Figure 5, the authors claim that dataset A1 would not yield good results with an unsupervised approach because the clusters overlap. I think that this would be strobgly dependent on i) the number of PC used, and ii) the explained variance of the chosen number of PC. Could the authors show how the clusters look using 3 PCs? What about more PCs? Would a clustering method correctly find the two units if, for example, 5 dimensions were used?

We thank the reviewer for this valuable suggestion. In response, we have conducted additional analyses to address these questions:

1. We had previously performed k-means clustering (using the known number of tracks/fibers) across all feature sets and datasets, evaluating clustering performance using V-measure, Adjusted Rand Index (ARI), and Normalized Mutual Information (NMI). In the revised manuscript, we have now included these results in the Supporting Information (Files S7-S9), allowing a comprehensive comparison across datasets and feature sets.

2. Exemplary for datasets A1 and A3, we have extended the manuscript as follows:

a. We present 2D and 3D scatter plots based on the first 2 and 3 principal components, respectively, to visually assess the cluster overlap (see Figures 5A-B, 5D-E).

b. For A1, we performed k-means clustering using 2-8 principal components and computed V-measure, ARI, and NMI scores at each dimensionality. These results are summarized in a new table (see Table 2).

c. The results show that increasing the number of PCs from 2 to 8 unfortunately does not substantially improve clustering performance with low scores (V-measures 0.59-0.62). This suggests that even with more PCs, unsupervised methods struggle to separate the overlapping units.

3. Regarding the explained variance, we computed the cumulative explained variance for the chosen number of PCs (2-8) for datasets A1 and A3 (Supporting Information S6). Despite capturing a large fraction of the variance, the cluster separability remains poor, reinforcing our original claim that overlapping feature distributions limit the effectiveness of unsupervised approaches in the frequently used PCA feature space.

In the same figure, I found panels C-D quite puzzling. The blue and green sub-clusters that the authors state are from the same unit will likely have very different waveforms/templates? How do the authors explain this? Is this something commonly observed in spikes from C-Nociceptors? Could it be that two different fibers are elicited from the same stimulus and have the same latency?

We appreciate the reviewer bringing this issue to our attention. Upon closer inspection of panels C and D, we identified an alignment error that artificially separated the green sub-cluster. We sincerely apologize for this mistake. In the revised manuscript, we have removed panels C and D from the figure. We have recomputed all results and carefully checked the spike alignment in all datasets.

We would additionally like to address the problem of the potential spike overlap, as it is a well-known challenge in the spike sorting field. In theory, two different fibers could indeed be activated by the same stimulus and have identical conduction latencies, potentially leading to superimposed spike shapes. However, such occurrences are extremely rare. Only in the case when two nerve fibers have exactly the same latency, and when sometimes one fiber is not activated by the stimulus, the recorded waveform may be shaped by the spike of a single axon, resulting in a slight change in spike shape. This is excluded by visual inspection using the marking method (see Figure 6B). We apply different protocols, such as mechanical stimulation, and can observe different amounts of activity-dependent conduction velocity slowing. Differences in the amount of slowing would reveal distinct fibers even if they have the same initial latency.

2. In cases where the waveforms/templates are very similar (e.g., Figure 3B-D), is it possible that an individual stimulus results in a fiber firing twice, in sequence? With the marking methods, these two spikes would be classified as separate units, and as the authors suggest, it would be virtually impossible to spike sort them accurately. On the other hand, isn't it unlikely, given the geometry of the fibers and the relative position to the electrode, that the spikes come indeed from the same fiber?

We agree that, in principle, it is possible for a peripheral axon to conduct two or more action potentials elicited by the same stimulus at different branches within the skin (cf. Weidner et al., 2000). However, in healthy volunteers, this phenomenon is rare. At branching points, the fastest action potential typically enters the branches retrogradely and collides with anterogradely conducted action potentials. The occurrence of this so-called “unidirectional conduction block” leading to double, triple, or multiple spiking is recognized by a progressive activity-dependent slowing of conduction velocity, without the application of additional stimuli.

Here, there is no evidence of multiple spiking within a single axon. Rather, the recorded action potentials originate from two distinct axons. It is a well-known observation across multiple laboratories that, in C-fiber recordings, when using a single extracellular electrode, the shape of action potentials from different axons can appear very similar.

3. Figure 4A. the authors restrict this similarity analysis to datasets with 2 tracks. I think that the number of data points could be extended also to multi-track dataset by considering, for each unit, a second unit with the lowest similaroty (worst case scenario).

In practical applications of C-fiber microneurography, we typically focus on the best-isolated track, especially when the goal is to identify well-separated fibers for further analysis. Including the most similar (i.e., least distinct) pairings from two tracks may not be a fair comparison. However, to address the reviewer’s point and ensure transparency, we have included these values for multi-track datasets as grey crosses in Figure 4A. This allows readers to observe the range of distances compared to the maximal accuracy while maintaining the focus of our main analysis.

4. Since the authors suggest that the method could be applied to spontaneous spikes as well, it would be nice to see that in action. How many spontaneous spikes could be classified on the used datasets?

It is important to note that peripheral afferent C-nociceptors in healthy volunteers are typically not spontaneously active, as their primary role is to signal potentially tissue-damaging stimuli. To address this limitation and explore the classifier's applicability to non-tracked activity, we plan to adjust our current stimulation protocol based on the marking method and use additional electrical-induced activity in future work. Such stimulation provides trackable, ground truth-labeled spikes and can serve as a controlled model for spontaneous firing. In the next steps, we aim to extend this work to include spikes evoked by mechanical, thermal, or chemical stimuli, which model in many respects spontaneous activity. To provide clarity, we have updated the Introduction, Discussion, and Conclusions and Outlook sections to emphasize that these will be the future steps in our research.

Minor points:

1. L142: SS-SPDF: first time this acronym is used. May be worth spelling it out

Thank you for pointing this out. We have now spelled out the acronym upon its first mention to improve clarity for the reader.

2. L352: why do the authors state that the study is about "spike morphology"? I would classify it as spike sorting analysis only

Thank you for the comment. While our work indeed involves spike sorting analysis, we also aimed to provide a broader, systematic overview of spike morphology in recordings made with microneurography. To our knowledge, this is the first comprehensive analysis focused on characterizing the diversity and structure of spike shapes recorded with this method.

To support this, we have added representative templates from the analyzed recordings to the Supporting Information to highlight the morphological variety observed (S5). We have revised the Introduction (see lines 136-150) to clarify that a central aim of the study was to evaluate spike sorting performance, and additionally, we systematically assess and summarize spike morphology characteristics across datasets.

3. L376-378: what other applications could it be transfered to?

Thank you for the question. We have revised the Conclusions and Outlook (lines 545-553) section to be more specific about potential applications. In particular, we included a short section outlining example scenarios where our approach could be transferred, such as other peripheral recordings from sympathetic C-fibers, which are important for heart/blood pressure control and thus for growing number of cardiovascular diseases or sensory A-fibers, whose input is important for motor control and in this line important for prosthesis development.

4. Table2: I would rename the Track*, R/S/T, Unit* consistently. The different naming makes it more confusing. You could just use Track1/2/3 or Unit1/2/3

Thank you for the suggestion, we have changed the names of the tracks to avoid confusion.

5. L481: Panda's -> pandas

Thank you, we have changed it.

6. Table 3: since most of steps are the same, a diagram/flowchart would be easier to understand. The differences are minor and could be explained in the text/captipn

Thank you for the helpful suggestion. We agree that a visual representation improves clarity. We have replaced Table 3 with a flowchart (now Fig 7) illustrating the main processing steps and moved the minor differences between pipelines to the text.

7. L518-519: this width is usually referred to as FWHM (full-width half maximum)

Thank you for pointing this out. We have updated the terminology in the manuscript to refer to the width as FWHM (full-width at half maximum), as suggested.

8. L558: What is W2? I guess that Wraw is the entire waveform (30-90 points), but W2 is not specified

We apologize for the confusion. W2 describes the principal components with two dimensions of the entire waveform and is defined in Table 5, where we provide a description of all feature sets. We have renamed W2 to W2-PCA for more clarity. The explanation can be found in the Materials and Methods section (beginning in line 762).

9. L569: where does the 23 come from? Aren't the new features 2 (S2/S3) instead of 4?

The original feature vector derived from the SS-SPDF method consisted of 24 features. However, we had to remove feature 4 as described in the Materials and Methods section, resulting in a 23-dimensional feature vector. Additionally, we have removed the PCA step, as suggested by Reviewer 3, P1, after realizing it did not contribute to redundancy reduction or performance improvement. We have revised the relevant section in the manuscript.

Reviewer #2: In this manuscript, the authors describe and evaluate the feasibility of a novel approach to sort spikes in electrophysiological recordings from nociceptive C-fibers in the peripheral nervous system, obtained using microneurography. The approach involves using SVM for the supervised classification of previously labeled spike data, opposed to existing unsupervised methods. The labelling of spikes is performed by employing the marking method, which leverages the difference in conduction velocities between fibers recorded simultaneously by an electrode and the activity-dependent slowing phenomenon in C-fibers. In the marking method, action potentials belonging to a single C-fiber are time-locked to a background low-frequency stimulus. By visualizing the recordings after several stimulation pulses in parallel, spikes originating from the same fiber will apear aligned in time (named a “track”), allowing to assign each spike to an individual fiber, and to extract the spike waveforms from the raw data. From the spike waveform data, the authors extract distinct feature sets (simple features, SS-SPDF features, and raw waveform features), which are the inputs to the SVM classifier. The SVM training/testing is performed for each recorded dataset (26 in total, from two different labs) using 5-fold cross validation, allowing to evaluate the sorting peformance of each feature set on a per-dataset basis, which could be used to guide the selection of the best features for sorting in a given dataset. In the end, the authors analyze and discuss the classification performance of each feature set according to general data features, such as number of individual fibers recorded and similarity of spike waveforms across the fibers in a single recording. They also highglight how the supervised method can be more efficient to sort spikes from different fibers compared to methods based on PCA followed by unsupervised clustering.

The manuscript is relevant for research involving microneurography and may advance the analysis of data from recordings of C-fibers. Moreover, the authors aimed to make the software code to apply the proposed method available, and describe efforts to make the analysis pipeline usable with different proprietary input file formats (leveraging open standards for data and metadata such as odML and NIX). However, there are points that need improvement and clarification in the manuscript.

We thank the reviewer for the constructive feedback on both the manuscript and the accompanying code. We appreciate the recognition of the manuscript’s relevance to microneurography research and its potential contribution to the analysis of C-fiber recordings. We are especially grateful for the suggestion to restructure the code using Snakemake. This recommendation significantly improved the efficiency and usability of our pipeline and greatly simplified the workflow for us.

1. The whole analysis rely on labeled data for ground truth using the marking method, but the manuscript does not detail how the spike tracks were actually obtained. Although the authors took their time to explain the rationale of the method, with helpful illustrative figures, the section in the methods (lines 448-472) does not detail how the actual procedure is executed with the raw data obtained in a recording from Aachen or Bristol. In the main text, the authors just describe this as a pre-analysis (line 165). In addition, there is no code in the GitHub repository for obtaining that track information (assumed to be stored in a NIX file after initial curation; Table 3). Therefore, the manuscript must present a more detailed description on how the marking method is performed as one part of the whole pipeline to obtain the starting spike data for the classification (e.g., software tools, manual curation steps, detailed signal preprocessing steps and parameters). The code should also be sh

---

## [Decision Letter · Decision Letter 1]

18 Jul 2025

Supervised Spike Sorting Feasibility of Noisy Single-Electrode Extracellular Recordings: Systematic Study of Human C-Nociceptors recorded via Microneurography

PONE-D-25-04960R1

Dear Dr. Troglio,

I am pleased to inform you that your manuscript has been recommened for publication by all three reviewers. It will be formally accepted once it meets any outstanding technical requirements.

Within one week, you will receive an e-mail detailing the required amendments. When these have been addressed, you will receive a formal acceptance letter and your manuscript will be scheduled for publication.

If your institution or institutions have a press office, please notify them about your upcoming paper to help maximize its impact. If they will be preparing press materials, please inform our press team as soon as possible -- no later than 48 hours after receiving the formal acceptance. Your manuscript will remain under strict press embargo until 2 pm Eastern Time on the date of publication. For more information, please contact onepress@plos.org.

Sincerely,

Nicholas V Swindale

Academic Editor

PLOS ONE

Additional Editor Comments (optional):

Reviewers' comments:

Reviewer's Responses to Questions

**Comments to the Author**

1. If the authors have adequately addressed your comments raised in a previous round of review and you feel that this manuscript is now acceptable for publication, you may indicate that here to bypass the “Comments to the Author” section, enter your conflict of interest statement in the “Confidential to Editor” section, and submit your "Accept" recommendation.

Reviewer #1: All comments have been addressed

Reviewer #2: All comments have been addressed

Reviewer #3: All comments have been addressed

2. Is the manuscript technically sound, and do the data support the conclusions?

Reviewer #1: Yes

Reviewer #2: Yes

Reviewer #3: Yes

3. Has the statistical analysis been performed appropriately and rigorously? 

Reviewer #1: Yes

Reviewer #2: Yes

Reviewer #3: Yes

4. Have the authors made all data underlying the findings in their manuscript fully available?

Reviewer #1: Yes

Reviewer #2: Yes

Reviewer #3: Yes

5. Is the manuscript presented in an intelligible fashion and written in standard English?

Reviewer #1: Yes

Reviewer #2: Yes

Reviewer #3: Yes

6. Review Comments to the Author

Reviewer #1: The authors addressed all my comments. I am glad that one of my comments allowed the authors to spot a bug in the code.

Reviewer #2: I thank the authors for the time to revise the manuscript, that is greatly improved. The authors performed a thorough rewriting of the text, to make their motivation, goals, and methods clear. The added comparison to other spike sorters strengthened the previous findings. The revised discussion now critically assessses the findings and limitations. The shared code was also refactored to use a WMS facilitating its use by the community. The open-source pipeline is easily applied to a new dataset, and the runs are more reproducible. All the concerns raised were addressed or adequately clarified in the rebuttal letter. The manuscript is suitable for publication.

Reviewer #3: FINAL REVIEW REPORT: Reviewer # 3

The authors have answered all my concerns and questions appropriately. In the new version of this manuscript, the authors have substantially improved the focus of the article to meet the demands of an audience more interested in experimental applications of this spike sorting method, mainly for human multi-unit electrophysiological extracellular nerve recordings (raw microneurography data).

Also, they have improved the methodological description, addressing all my major concerns about the alignment of the spikes, clustering, classification and curation processes. This version's workflow for spike sorting is more solid and a much better fit for the proposal.

In particular, the comparative aspects between the proposed method/algorithm and other approaches for spike sorting such as SS-SPDF method with Template Matching in spike phase-space, SpikeInterface and Support Vector Machines were appropriately addressed in the new version of the manuscript.

In addition, all my minor comments were also addressed.

I consider this type of mixed approach that integrating supervised spike-sorting with machine-learning models for C-nociceptor microneurography data very interesting and this is irrefutably necessary in clinical neurophysiology.

In my opinion, the manuscript has significantly improved with the suggestions and comments of all the reviewers, and with the new changes introduced by the authors. I have no futher suggestions, therefore, I accept this manuscript for publication in its current version.

Interesting paper - Job well done!!!

7. PLOS authors have the option to publish the peer review history of their article (what does this mean? ). If published, this will include your full peer review and any attached files.

**Do you want your identity to be public for this peer review?** For information about this choice, including consent withdrawal, please see our Privacy Policy .

Reviewer #1: **Yes: ** Alessio Paolo Buccino

Reviewer #2: No

Reviewer #3: No

---

## [Editor Report · Acceptance letter]

PONE-D-25-04960R1

PLOS ONE

Dear Dr. Troglio,

I'm pleased to inform you that your manuscript has been deemed suitable for publication in PLOS ONE. Congratulations! Your manuscript is now being handed over to our production team.

Kind regards,

on behalf of

Dr. Nicholas V Swindale

Academic Editor

PLOS ONE